# CROSSFORMER: TRANSFORMER UTILIZING CROSS-DIMENSION DEPENDENCY FOR MULTIVARIATE TIME SERIES FORECASTING

**Yunhao Zhang & Junchi Yan***
MoE Key Lab of Artificial Intelligence, Shanghai Jiao Tong University and Shanghai AI Lab
{zhangyunhao, yanjunchi}@sjtu.edu.cn
Code: https://github.com/Thinklab-SJTU/Crossformer

## ABSTRACT

Recently many deep models have been proposed for multivariate time series (MTS) forecasting. In particular, Transformer-based models have shown great potential because they can capture long-term dependency. However, existing Transformer-based models mainly focus on modeling the temporal dependency (cross-time dependency) yet often omit the dependency among different variables (cross-dimension dependency), which is critical for MTS forecasting. To fill the gap, we propose Crossformer, a Transformer-based model utilizing cross-dimension dependency for MTS forecasting. In Crossformer, the input MTS is embedded into a 2D vector array through the Dimension-Segment-Wise (DSW) embedding to preserve time and dimension information. Then the Two-Stage Attention (TSA) layer is proposed to efficiently capture the cross-time and cross-dimension dependency. Utilizing DSW embedding and TSA layer, Crossformer establishes a Hierarchical Encoder-Decoder (HED) to use the information at different scales for the final forecasting. Extensive experimental results on six real-world datasets show the effectiveness of Crossformer against previous state-of-the-arts.

## 1 INTRODUCTION

Multivariate time series (MTS) are time series with multiple dimensions, where each dimension represents a specific univariate time series (e.g. a climate feature of weather). MTS forecasting aims to forecast the future value of MTS using their historical values. MTS forecasting benefits the decision-making of downstream tasks and is widely used in many fields including weather (Angryk et al., 2020), energy (Demirel et al., 2012), finance (Patton, 2013), etc. With the development of deep learning, many models have been proposed and achieved superior performances in MTS forecasting (Lea et al., 2017; Qin et al., 2017; Flunkert et al., 2017; Rangapuram et al., 2018; Li et al., 2019a; Wu et al., 2020; Li et al., 2021). Among them, the recent Transformer-based models (Li et al., 2019b; Zhou et al., 2021; Wu et al., 2021a; Liu et al., 2021a; Zhou et al., 2022; Chen et al., 2022) show great potential thanks to their ability to capture long-term temporal dependency (cross-time dependency).

Besides cross-time dependency, the cross-dimension dependency is also critical for MTS forecasting, i.e. for a specific dimension, information from associated series in other dimensions may improve prediction. For example, when predicting future temperature, not only the historical temperature, but also historical wind speed helps to forecast. Some previous neural models explicitly capture the cross-dimension dependency, i.e. preserving the information of dimensions in the latent feature space and using convolution neural network (CNN) (Lai et al., 2018) or graph neural network (GNN) (Wu et al., 2020; Cao et al., 2020) to capture their dependency. However, recent Transformer-based models only implicitly utilize this dependency by embedding. In general, Transformer-based models embed data points in all dimensions at the same time step into a feature vector and try to capture dependency among different time steps (like Fig. 1 (b)). In this way, cross-time dependency is well captured, but cross-dimension dependency is not, which may limit their forecasting capability.

---

*Junchi Yan is the correspondence author. This work was in part supported by NSFC (61972250, U19B2035, 62222607) and Shanghai Municipal Science and Technology Major Project (2021SHZDZX0102).

To fill the gap, we propose Crossformer, a Transformer-based model that explicitly utilizes cross-dimension dependency for MTS forecasting. Specifically, we devise Dimension-Segment-Wise (DSW) embedding to process the historical time series. In DSW embedding, the series in each dimension is first partitioned into segments and then embedded into feature vectors. The output of DSW embedding is a 2D vector array where the two axes correspond to time and dimension. Then we propose the Two-Stage-Attention (TSA) layer to efficiently capture the cross-time and cross-dimension dependency among the 2D vector array. Using DSW embedding and TSA layer, Crossformer establishes a Hierarchical Encoder-Decoder (HED) for forecasting. In HED, each layer corresponds to a scale. The encoder's upper layer merges adjacent segments output by the lower layer to capture the dependency at a coarser scale. Decoder layers generate predictions at different scales and add them up as the final prediction. **The contributions of this paper are:**

**1)** We dive into the existing Transformer-based models for MTS forecasting and figure out that the cross-dimension dependency is not well utilized: these models simply embed data points of all dimensions at a specific time step into a single vector and focus on capturing the cross-time dependency among different time steps. Without adequate and explicit mining and utilization of cross-dimension dependency, their forecasting capability is empirically shown limited.

**2)** We develop Crossformer, a Transformer model utilizing cross-dimension dependency for MTS forecasting. This is one of the few transformer models (perhaps the first to our best knowledge) that *explicitly* explores and utilizes cross-dimension dependency for MTS forecasting.

**3)** Extensive experimental results on six real-world benchmarks show the effectiveness of our Crossformer against previous state-of-the-arts. Specifically, Crossformer ranks top-1 among the 9 models for comparison on 36 out of the 58 settings of varying prediction lengths and metrics and ranks top-2 on 51 settings.

## 2 RELATED WORKS

**Multivariate Time Series Forecasting.** MTS forecasting models can be roughly divided into statistical and neural models. Vector auto-regressive (VAR) model (Kilian & LÃtkepohl, 2017) and Vector auto-regressive moving average (VARMA) are typical statistical models, which assume linear cross-dimension and cross-time dependency. With the development of deep learning, many neural models have been proposed and often empirically show better performance than statistical ones. TCN (Lea et al., 2017) and DeepAR (Flunkert et al., 2017) treat the MTS data as a sequence of vectors and use CNN/RNN to capture the temporal dependency. LSTnet (Lai et al., 2018) employs CNN to capture cross-dimension dependency and RNN for cross-time dependency. Another category of works use graph neural networks (GNNs) to capture the cross-dimension dependency explicitly for forecasting (Li et al., 2018; Yu et al., 2018; Cao et al., 2020; Wu et al., 2020). For example, MTGNN (Wu et al., 2020) uses temporal convolution and graph convolution layers to capture cross-time and cross-dimension dependency. These neural models capture the cross-time dependency through CNN or RNN, which have difficulty in modeling long-term dependency.

**Transformers for MTS Forecasting.** Transformers (Vaswani et al., 2017) have achieved success in natural language processing (NLP) (Devlin et al., 2019), vision (CV) (Dosovitskiy et al., 2021) and speech processing (Dong et al., 2018). Recently, many Transformer-based models have been proposed for MTS forecasting and show great potential (Li et al., 2019b; Zhou et al., 2021; Wu et al., 2021a; Liu et al., 2021a; Zhou et al., 2022; Du et al., 2022). LogTrans (Li et al., 2019b) proposes the LogSparse attention that reduces the computation complexity of Transformer from $O(L^2)$ to $O\left(L(\log L)^2\right)$[1]. Informer (Zhou et al., 2021) utilizes the sparsity of attention score through KL divergence estimation and proposes ProbSparse self-attention which achieves $O(L \log L)$ complexity. Autoformer (Wu et al., 2021a) introduces a decomposition architecture with an Auto-Correlation mechanism to Transformer, which also achieves the $O(L \log L)$ complexity. Pyraformer (Liu et al., 2021a) introduces a pyramidal attention module that summarizes features at different resolutions and models the temporal dependencies of different ranges with the complexity of $O(L)$. FEDformer (Zhou et al., 2022) proposes that time series have a sparse representation in frequency domain and develop a frequency enhanced Transformer with the $O(L)$ complexity. Preformer (Du et al., 2022) divides the embedded feature vector sequence into segments and utilizes segment-wise correlation-based attention

---

[1]In this paper, the meaning of complexity refers to both time and space overhead and they are often the same. This is also in line with the presentation in existing works (Wu et al., 2021a; Liu et al., 2021a).

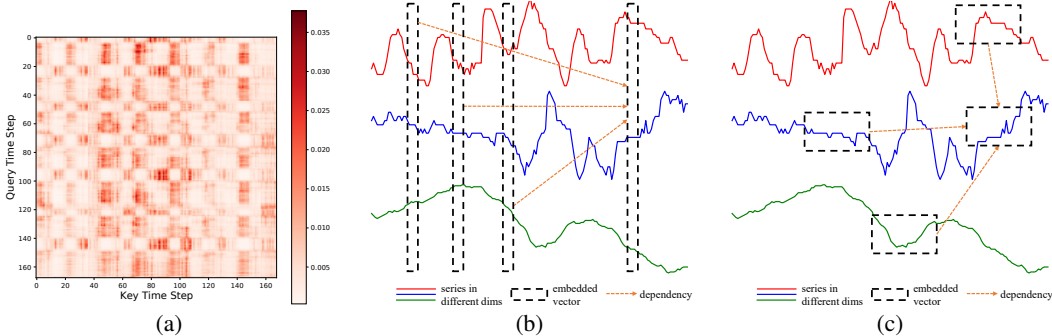

Figure 1: Illustration for our DSW embedding. (a) Self-attention scores from a 2-layer Transformer trained on ETTh1, showing that MTS data tends to be segmented. (b) Embedding method of previous Transformer-based models (Li et al., 2019b; Zhou et al., 2021; Wu et al., 2021a; Liu et al., 2021a): data points in different dimensions at the same step are embedded into a vector. (c) DSW embedding of Crossformer: in each dimension, nearby points over time form a segment for embedding.

for forecasting. These models mainly focus on reducing the complexity of cross-time dependency modeling, but omits the cross-dimension dependency which is critical for MTS forecasting.

**Vision Transformers.** Transformer is initially applied to NLP for sequence modeling, recent works apply transformer to CV tasks to process images (Dosovitskiy et al., 2021; Touvron et al., 2021; Liu et al., 2021b; Chen et al., 2021; Han et al., 2021). These works achieve state-of-the-art performance on various tasks in CV and inspire our work. ViT (Dosovitskiy et al., 2021) is one of the pioneers of vision transformers. The basic idea of ViT is to split an image into non-overlapping medium-sized patches, then it rearranges these patches into a sequence to be input to the Transformer. The idea of partitioning images into patches inspires our DSW embedding where MTS is split into dimension-wise segments. Swin Transformer (Liu et al., 2021b) performs local attention within a window to reduce the complexity and builds hierarchical feature maps by merging image patches. Readers can refer to the recent survey (Han et al., 2022) for comprehensive study on vision transformers.

## 3 METHODOLOGY

In multivariate time series forecasting, one aims to predict the future value of time series $\mathbf{x}_{T+1:T+\tau} \in \mathbb{R}^{\tau \times D}$ given the history $\mathbf{x}_{1:T} \in \mathbb{R}^{T \times D}$, where $\tau$, $T$ is the number of time steps in the future and past, respectively[2]. $D > 1$ is the number of dimensions. A natural assumption is that these $D$ series are associated (e.g. climate features of weather), which helps to improve the forecasting accuracy. To utilize the cross-dimension dependency, in Section 3.1, we embed the MTS using Dimension-Segment-Wise (DSW) embedding. In Section 3.2, we propose a Two-Stage Attention (TSA) layer to efficiently capture the dependency among the embedded segments. In Section 3.3, using DSW embedding and TSA layer, we construct a hierarchical encoder-decoder (HED) to utilize information at different scales for final forecasting.

### 3.1 DIMENSION-SEGMENT-WISE EMBEDDING

To motivate our approach, we first analyze the embedding methods of the previous Transformer-based models for MTS forecasting (Zhou et al., 2021; Wu et al., 2021a; Liu et al., 2021a; Zhou et al., 2022). As shown in Fig. 1 (b), existing methods embed data points at the same time step into a vector: $\mathbf{x}_t \to \mathbf{h}_t, \mathbf{x}_t \in \mathbb{R}^D, \mathbf{h}_t \in \mathbb{R}^{d_{model}}$, where $\mathbf{x}_t$ represents all the data points in $D$ dimensions at step $t$. In this way, the input $\mathbf{x}_{1:T}$ is embedded into $T$ vectors $\{\mathbf{h}_1, \mathbf{h}_2, \dots, \mathbf{h}_T\}$. Then the dependency among the $T$ vectors is captured for forecasting. Therefore, previous Transformer-based models mainly capture cross-time dependency, while the cross-dimension dependency is not explicitly captured during embedding, which limits their forecasting capability.

Transformer was originally developed for NLP (Vaswani et al., 2017), where each embedded vector represents an informative word. For MTS, a single value at a step alone provides little information.

---

[2]In this work, we mainly focus on forecasting using only past data without covariates. But covariates can be easily incorporated in Crossformer through embedding. Details are shown in Appendix D.2.

While it forms informative pattern with nearby values in time domain. Fig. 1 (a) shows a typical attention score map of original Transformer for MTS forecasting. We can see that attention values have a tendency to segment, i.e. close data points have similar attention weights.

Based on the above two points, we argue that an embedded vector should represent a series segment of single dimension (Fig. 1 (c)), rather than the values of all dimensions at single step (Fig. 1 (b)). To this end, we propose Dimension-Segment-Wise (DSW) embedding where the points in each dimension are divided into segments of length $L_{seg}$ and then embedded:

$$
\begin{aligned}
\mathbf{x}_{1:T} &= \left\{ \mathbf{x}_{i,d}^{(s)} \big| 1 \le i \le \frac{T}{L_{seg}}, 1 \le d \le D \right\} \\
\mathbf{x}_{i,d}^{(s)} &= \left\{ x_{t,d} \big| (i-1) \times L_{seg} < t \le i \times L_{seg} \right\}
\end{aligned}
\tag{1}
$$

where $\mathbf{x}_{i,d}^{(s)} \in \mathbb{R}^{L_{seg}}$ is the $i$-th segment in dimension $d$ with length $L_{seg}$. For convenience, we assume that $T, \tau$ are divisible by $L_{seg}$[3]. Then each segment is embedded into a vector using linear projection added with a position embedding:

$$
\mathbf{h}_{i,d} = \mathbf{E}\mathbf{x}_{i,d}^{(s)} + \mathbf{E}_{i,d}^{(pos)}
\tag{2}
$$

where $\mathbf{E} \in \mathbb{R}^{d_{model} \times L_{seg}}$ denotes the learnable projection matrix, and $\mathbf{E}_{i,d}^{(pos)} \in \mathbb{R}^{d_{model}}$ denotes the learnable position embedding for position $(i,d)$. After embedding, we obtain a 2D vector array $\mathbf{H} = \left\{ \mathbf{h}_{i,d} \big| 1 \le i \le \frac{T}{L_{seg}}, 1 \le d \le D \right\}$, where each $\mathbf{h}_{i,d}$ represents a univariate time series segment. The idea of segmentation is also used in Du et al. (2022), which splits the embedded 1D vector sequence into segments to compute the Segment-Correlation in order to enhance locality and reduce computation complexity. However, like other Transformers for MTS forecasting, it does not explicitly capture cross-dimension dependency.

## 3.2 Two-Stage Attention Layer

For the obtained 2D array $\mathbf{H}$, one can flatten it into a 1D sequence so that it can be input to a canonical Transformer like ViT (Dosovitskiy et al., 2021) does in vision. While we have specific considerations: 1) Different from images where the axes of height and width are interchangeable, the axes of time and dimension for MTS have different meanings and thus should be treated differently. 2) Directly applying self-attention on 2D array will cause the complexity of $O(D^2 \frac{T^2}{L_{seg}^2})$, which is unaffordable for large $D$. Therefore, we propose the Two-Stage Attention (TSA) Layer to capture cross-time and cross-dimension dependency among the 2D vector array, as sketched in Fig. 2 (a).

**Cross-Time Stage**  Given a 2D array $\mathbf{Z} \in \mathbb{R}^{L \times D \times d_{model}}$ as the input of the TSA Layer, where $L$ and $D$ are the number of segments and dimensions, respectively. $\mathbf{Z}$ here can be the output of DSW embedding or lower TSA layers. For convenience, in the following, we use $\mathbf{Z}_{i,:}$ to denote the vectors of all dimensions at time step $i$, $\mathbf{Z}_{:,d}$ for those of all time steps in dimension $d$. In the cross-time stage, we directly apply multi-head self-attention (MSA) to each dimension:

$$
\begin{aligned}
\hat{\mathbf{Z}}_{:,d}^{time} &= \texttt{LayerNorm}\Big( \mathbf{Z}_{:,d} + \texttt{MSA}^{time}(\mathbf{Z}_{:,d}, \mathbf{Z}_{:,d}, \mathbf{Z}_{:,d}) \Big) \\
\mathbf{Z}^{time} &= \texttt{LayerNorm}\Big( \hat{\mathbf{Z}}^{time} + \texttt{MLP}(\hat{\mathbf{Z}}^{time}) \Big)
\end{aligned}
\tag{3}
$$

where $1 \le d \le D$ and $\texttt{LayerNorm}$ denotes layer normalization as widely adopted in Vaswani et al. (2017); Dosovitskiy et al. (2021); Zhou et al. (2021), $\texttt{MLP}$ denotes a multi-layer (two in this paper) feedforward network, $\texttt{MSA}(\mathbf{Q}, \mathbf{K}, \mathbf{V})$ denotes the multi-head self-attention (Vaswani et al., 2017) layer where $\mathbf{Q}, \mathbf{K}, \mathbf{V}$ serve as queries, keys and values. All dimensions ($1 \le d \le D$) share the same MSA layer. $\hat{\mathbf{Z}}^{time}, \mathbf{Z}^{time}$ denotes the output of the MSA and MLP.

The computation complexity of cross-time stage is $O(DL^2)$. After this stage, the dependency among time segments in the same dimension is captured in $\mathbf{Z}^{time}$. Then $\mathbf{Z}^{time}$ becomes the input of Cross-Dimension Stage to capture cross-dimension dependency.

---

[3]If not, we pad them to proper length. See details in Appendix D.1.

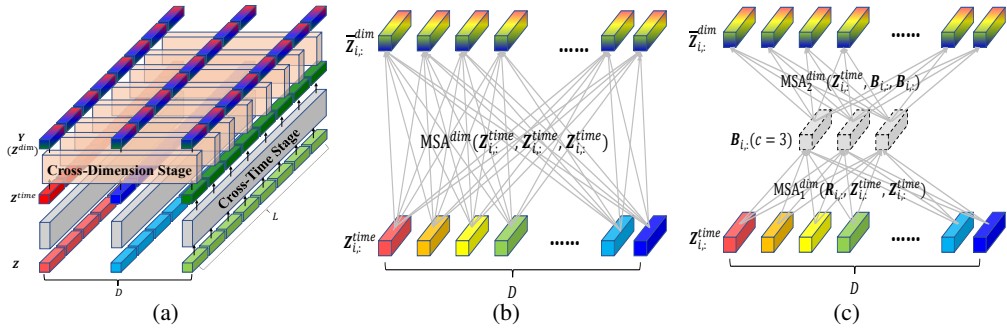

Figure 2: The TSA layer. (a) Two-Stage Attention Layer to process a 2D vector array representing multivariate time series: each vector refers to a segment of the original series. The whole vector array goes through the Cross-Time Stage and Cross-Dimension Stage to get corresponding dependency. (b) Directly using MSA in Cross-Dimension Stage to build the $D$-to-$D$ connection results in $O(D^2)$ complexity. (c) Router mechanism for Cross-Dimension Stage: a small fixed number ($c$) of "routers" gather information from all dimensions and then distribute the gathered information. The complexity is reduced to $O(2cD) = O(D)$.

**Cross-Dimension Stage**    We can use a large $L_{seg}$ for long sequence in DSW Embedding to reduce the number of segments $L$ in cross-time stage. While in Cross-Dimension Stage, we can not partition dimensions and directly apply MSA will cause the complexity of $O(D^2)$ (as shown in Fig. 2 (b)), which is unaffordable for datasets with large $D$. Instead, we propose the router mechanism for potentially large $D$. As shown in Fig. 2 (c), we set a small fixed number ($c << D$) of learnable vectors for each time step $i$ as routers. These routers first aggregate messages from all dimensions by using routers as query in MSA and vectors of all dimensions as key and value. Then routers distribute the received messages among dimensions by using vectors of dimensions as query and aggregated messages as key and value. In this way, the all-to-all connection among $D$ dimensions are built:

$$\mathbf{B}_{i,:} = \text{MSA}_1^{dim}(\mathbf{R}_{i,:}, \mathbf{Z}_{i,:}^{time}, \mathbf{Z}_{i,:}^{time}), 1 \leq i \leq L$$
$$\overline{\mathbf{Z}}_{i,:}^{dim} = \text{MSA}_2^{dim}(\mathbf{Z}_{i,:}^{time}, \mathbf{B}_{i,:}, \mathbf{B}_{i,:}), 1 \leq i \leq L$$
$$\hat{\mathbf{Z}}^{dim} = \text{LayerNorm}\left(\mathbf{Z}^{time} + \overline{\mathbf{Z}}^{dim}\right) \tag{4}$$
$$\mathbf{Z}^{dim} = \text{LayerNorm}\left(\hat{\mathbf{Z}}^{dim} + \text{MLP}(\hat{\mathbf{Z}}^{dim})\right)$$

where $\mathbf{R} \in \mathbb{R}^{L \times c \times d_{model}}$ ($c$ is a constant) is the learnable vector array serving as routers. $\mathbf{B} \in \mathbb{R}^{L \times c \times d_{model}}$ is the aggregated messages from all dimensions. $\overline{\mathbf{Z}}^{dim}$ denotes output of the router mechanism. All time steps ($1 \leq i \leq L$) share the same $\text{MSA}_1^{dim}, \text{MSA}_2^{dim}$. $\hat{\mathbf{Z}}^{dim}, \mathbf{Z}^{dim}$ denote output of skip connection and MLP respectively. The router mechanism reduce the complexity from $O(D^2 L)$ to $O(DL)$.

Adding up Eq. 3 and Eq. 4, we model the two stages as:

$$\mathbf{Y} = \mathbf{Z}^{dim} = \text{TSA}(\mathbf{Z}) \tag{5}$$

where $\mathbf{Z}, \mathbf{Y} \in \mathbb{R}^{L \times D \times d_{model}}$ denotes the input and output vector array of TSA layer, respectively. Note that the overall computation complexity of the

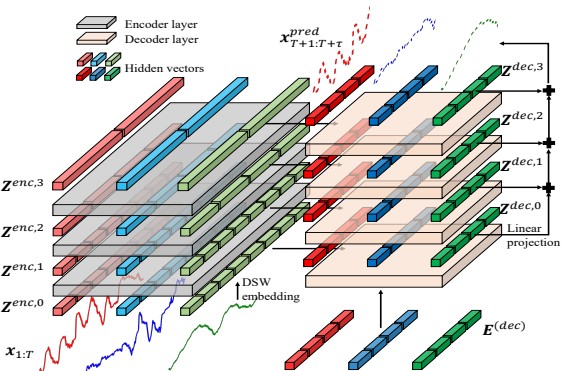

Figure 3: Architecture of the Hierarchical Encoder-Decoder in Crossformer with 3 encoder layers. The length of each vector denotes the covered time range. The encoder (left) uses TSA layer and segment merging to capture dependency at different scales: a vector in upper layer covers a longer range, resulting in dependency at a coarser scale. Exploring different scales, the decoder (right) makes the final prediction by forecasting at each scale and adding them up.

TSA layer is $O(DL^2 + DL) = O(DL^2)$. After the Cross-Time and Cross-Dimension Stages, every two segments (i.e. $\mathbf{Z}_{i_1,d_1}, \mathbf{Z}_{i_2,d_2}$) in $\mathbf{Z}$ are connected, as such both cross-time and cross-dimension dependencies are captured in $\mathbf{Y}$.

### 3.3 HIERARCHICAL ENCODER-DECODER

Hierarchical structures are widely used in Transformers for MTS forecasting to capture information at different scales (Zhou et al., 2021; Liu et al., 2021a). In this section, we use the proposed DSW embedding, TSA layer and segment merging to construct a Hierarchical Encoder-Decoder (HED). As shown in Fig. 3, the upper layer utilizes information at a coarser scale for forecasting. Forecasting values at different scales are added to output the final result.

**Encoder** In each layer of the encoder (except the first layer), every two adjacent vectors in time domain are merged to obtain the representation at a coarser level. Then a TSA layer is applied to capture dependency at this scale. This process is modeled as $\mathbf{Z}^{enc,l} = \text{Encoder}(\mathbf{Z}^{enc,l-1})$:

$$
\begin{cases}
l = 1: & \hat{\mathbf{Z}}^{enc,l} = \mathbf{H} \\
l > 1: & \hat{\mathbf{Z}}^{enc,l}_{i,d} = \mathbf{M}[\mathbf{Z}^{enc,l-1}_{2i-1,d} \cdot \mathbf{Z}^{enc,l-1}_{2i,d}], 1 \leq i \leq \frac{L_{l-1}}{2}, 1 \leq d \leq D
\end{cases}
\tag{6}
$$
$$
\mathbf{Z}^{enc,l} = \text{TSA}(\hat{\mathbf{Z}}^{enc,l})
$$

where $\mathbf{H}$ denotes the 2D array obtained by DSW embedding; $\mathbf{Z}^{enc,l}$ denotes the output of the $l$-th encoder layer; $\mathbf{M} \in \mathbb{R}^{d_{model} \times 2d_{model}}$ denotes a learnable matrix for segment merging; $[\cdot]$ denotes the concatenation operation; $L_{l-1}$ denotes the number of segments in each dimension in layer $l - 1$, if it is not divisible by 2, we pad $\mathbf{Z}^{enc,l-1}$ to the proper length; $\hat{\mathbf{Z}}^{enc,l}$ denotes the array after segment merging in the $i$-th layer. Suppose there are $N$ layers in the encoder, we use $\mathbf{Z}^{enc,0}, \mathbf{Z}^{enc,1}, \ldots, \mathbf{Z}^{enc,N}, (\mathbf{Z}^{enc,0} = \mathbf{H})$ to represent the $N + 1$ outputs of the encoder. The complexity of each encoder layer is $O(D\frac{T^2}{L_{seg}^2})$.

**Decoder** Obtaining the $N + 1$ feature arrays output by the encoder, we use $N + 1$ layers (indexed by $0, 1, \ldots, N$) in decoder for forecasting. Layer $l$ takes the $l$-th encoded array as input, then outputs a decoded 2D array of layer $l$. This process is summarized as $\mathbf{Z}^{dec,l} = \text{Decoder}(\mathbf{Z}^{dec,l-1}, \mathbf{Z}^{enc,l})$:

$$
\begin{cases}
l = 0: & \tilde{\mathbf{Z}}^{dec,l} = \text{TSA}(\mathbf{E}^{(dec)}) \\
l > 0: & \tilde{\mathbf{Z}}^{dec,l} = \text{TSA}(\mathbf{Z}^{dec,l-1})
\end{cases}
$$
$$
\overline{\mathbf{Z}}^{dec,l}_{:,d} = \text{MSA}\left(\tilde{\mathbf{Z}}^{dec,l}_{:,d}, \mathbf{Z}^{enc,l}_{:,d}, \mathbf{Z}^{enc,l}_{:,d}\right), 1 \leq d \leq D
\tag{7}
$$
$$
\hat{\mathbf{Z}}^{dec,l} = \text{LayerNorm}\left(\tilde{\mathbf{Z}}^{dec,l} + \overline{\mathbf{Z}}^{dec,l}\right)
$$
$$
\mathbf{Z}^{dec,l} = \text{LayerNorm}\left(\hat{\mathbf{Z}}^{dec,l} + \text{MLP}(\hat{\mathbf{Z}}^{dec,l})\right)
$$

where $\mathbf{E}^{(dec)} \in \mathbb{R}^{\frac{\tau}{L_{seg}} \times D \times d_{model}}$ denotes the learnable position embedding for decoder. $\tilde{\mathbf{Z}}^{dec,l}$ is the output of TSA. The MSA layer takes $\tilde{\mathbf{Z}}^{dec,l}_{:,d}$ as query and $\mathbf{Z}^{enc,l}_{:,d}$ as the key and value to build the connection between encoder and decoder. The output of MSA is denoted as $\overline{\mathbf{Z}}^{dec,l}_{:,d}$. $\hat{\mathbf{Z}}^{dec,l}, \mathbf{Z}^{dec,l}$ denote the output of skip connection and MLP respectively. We use $\mathbf{Z}^{dec,0}, \mathbf{Z}^{enc,1}, \ldots, \mathbf{Z}^{dec,N}$ to represent the decoder output. The complexity of each decoder layer is $O\left(D\frac{\tau(T+\tau)}{L_{seg}^2}\right)$.

Linear projection is applied to each layer's output to yield the prediction of this layer. Layer predictions are summed to make the final prediction (for $l = 0, \ldots, N$):

$$
\text{for } l = 0, \ldots, N: \mathbf{x}^{(s),l}_{i,d} = \mathbf{W}^l \mathbf{Z}^{dec,l}_{i,d} \qquad \mathbf{x}^{pred,l}_{T+1:T+\tau} = \left\{\mathbf{x}^{(s),l}_{i,d} | 1 \leq i \leq \frac{\tau}{L_{seg}}, 1 \leq d \leq D\right\}
\tag{8}
$$
$$
\mathbf{x}^{pred}_{T+1:T+\tau} = \sum_{l=0}^{N} \mathbf{x}^{pred,l}_{T+1:T+\tau}
$$

where $\mathbf{W}^l \in \mathbb{R}^{L_{seg} \times d_{model}}$ is a learnable matrix to project a vector to a time series segment. $\mathbf{x}^{(s),l}_{i,d} \in \mathbb{R}^{L_{seg}}$ denotes the $i$-th segment in dimension $d$ of the prediction. All the segments in layer

Table 1: MSE/MAE with different prediction lengths. Bold/underline indicates the best/second. Results of LSTMa, LSTnet, Transformer, Informer on the first 4 datasets are from Zhou et al. (2021).

| Models | | LSTMa | | LSTnet | | MTGNN | | Transformer | | Informer | | Autoformer | | Pyraformer | | FEDformer | | Crossformer | |
|---|---|---|---|---|---|---|---|---|---|---|---|---|---|---|---|---|---|---|---|
| Metric | | MSE | MAE | MSE | MAE | MSE | MAE | MSE | MAE | MSE | MAE | MSE | MAE | MSE | MAE | MSE | MAE | MSE | MAE |
| ETTh1 | 24 | 0.650 | 0.624 | 1.293 | 0.901 | 0.336 | 0.393 | 0.620 | 0.577 | 0.577 | 0.549 | 0.439 | 0.440 | 0.493 | 0.507 | 0.318 | 0.384 | **0.305** | **0.367** |
| | 48 | 0.720 | 0.675 | 1.456 | 0.960 | 0.386 | 0.429 | 0.692 | 0.671 | 0.685 | 0.625 | 0.429 | 0.442 | 0.554 | 0.544 | **0.342** | **0.396** | 0.352 | 0.394 |
| | 168 | 1.212 | 0.867 | 1.997 | 1.214 | 0.466 | 0.474 | 0.947 | 0.797 | 0.931 | 0.752 | 0.493 | 0.479 | 0.781 | 0.675 | 0.412 | 0.449 | 0.410 | 0.441 |
| | 336 | 1.424 | 0.994 | 2.655 | 1.369 | 0.736 | 0.643 | 1.094 | 0.813 | 1.128 | 0.873 | 0.509 | 0.492 | 0.912 | 0.747 | 0.456 | 0.474 | 0.440 | 0.461 |
| | 720 | 1.960 | 1.322 | 2.143 | 1.380 | 0.916 | 0.750 | 1.241 | 0.917 | 1.215 | 0.896 | 0.539 | 0.537 | 0.993 | 0.792 | 0.521 | **0.515** | 0.519 | 0.524 |
| ETTm1 | 24 | 0.621 | 0.629 | 1.968 | 1.170 | 0.260 | 0.324 | 0.306 | 0.371 | 0.323 | 0.369 | 0.410 | 0.428 | 0.310 | 0.371 | 0.290 | 0.364 | **0.211** | **0.293** |
| | 48 | 1.392 | 0.939 | 1.999 | 1.215 | 0.386 | 0.408 | 0.465 | 0.470 | 0.494 | 0.503 | 0.485 | 0.464 | 0.465 | 0.464 | 0.342 | 0.396 | **0.300** | **0.352** |
| | 96 | 1.339 | 0.913 | 2.762 | 1.542 | 0.428 | 0.446 | 0.681 | 0.612 | 0.678 | 0.614 | 0.502 | 0.476 | 0.520 | 0.504 | 0.366 | 0.412 | **0.320** | **0.373** |
| | 288 | 1.740 | 1.124 | 1.257 | 2.076 | 0.469 | 0.488 | 1.162 | 0.879 | 1.056 | 0.786 | 0.604 | 0.522 | 0.729 | 0.657 | **0.398** | **0.433** | 0.404 | 0.427 |
| | 672 | 2.736 | 1.555 | 1.917 | 2.941 | 0.620 | 0.571 | 1.231 | 1.103 | 1.192 | 0.926 | 0.607 | 0.530 | 0.980 | 0.678 | **0.455** | **0.464** | 0.569 | 0.528 |
| WTH | 24 | 0.546 | 0.570 | 0.615 | 0.545 | 0.307 | 0.356 | 0.349 | 0.397 | 0.335 | 0.381 | 0.363 | 0.396 | 0.301 | 0.359 | 0.357 | 0.412 | **0.294** | **0.343** |
| | 48 | 0.829 | 0.677 | 0.660 | 0.589 | 0.388 | 0.422 | 0.386 | 0.433 | 0.395 | 0.459 | 0.456 | 0.462 | 0.376 | 0.421 | 0.428 | 0.458 | **0.370** | **0.411** |
| | 168 | 1.038 | 0.835 | 0.748 | 0.647 | 0.498 | 0.512 | 0.613 | 0.582 | 0.608 | 0.567 | 0.574 | 0.548 | 0.519 | 0.521 | 0.564 | 0.541 | **0.473** | **0.494** |
| | 336 | 1.657 | 1.059 | 0.782 | 0.683 | 0.506 | 0.523 | 0.707 | 0.634 | 0.702 | 0.620 | 0.600 | 0.571 | 0.539 | 0.543 | 0.533 | 0.536 | **0.495** | **0.515** |
| | 720 | 1.536 | 1.109 | 0.851 | 0.757 | **0.510** | **0.527** | 0.834 | 0.741 | 0.831 | 0.731 | 0.587 | 0.570 | 0.547 | 0.553 | 0.562 | 0.557 | 0.526 | 0.542 |
| ECL | 48 | 0.486 | 0.572 | 0.369 | 0.445 | 0.173 | 0.280 | 0.334 | 0.399 | 0.344 | 0.393 | 0.241 | 0.351 | 0.478 | 0.471 | 0.229 | 0.338 | **0.156** | **0.255** |
| | 168 | 0.574 | 0.602 | 0.394 | 0.476 | 0.236 | 0.320 | 0.353 | 0.420 | 0.368 | 0.424 | 0.299 | 0.387 | 0.452 | 0.455 | 0.263 | 0.361 | **0.231** | **0.309** |
| | 336 | 0.886 | 0.795 | 0.419 | 0.477 | 0.328 | 0.373 | 0.381 | 0.439 | 0.381 | 0.431 | 0.375 | 0.428 | 0.463 | 0.456 | 0.305 | 0.386 | 0.323 | **0.369** |
| | 720 | 1.676 | 1.095 | 0.556 | 0.565 | 0.422 | **0.410** | 0.391 | 0.438 | 0.406 | 0.443 | 0.377 | 0.434 | 0.480 | 0.461 | **0.372** | 0.434 | 0.404 | 0.423 |
| | 960 | 1.591 | 1.128 | 0.605 | 0.599 | 0.471 | 0.451 | 0.492 | 0.550 | 0.460 | 0.548 | **0.366** | **0.426** | 0.550 | 0.489 | 0.393 | 0.449 | 0.433 | 0.438 |
| ILI | 24 | 4.220 | 1.335 | 4.975 | 1.660 | 4.265 | 1.387 | 3.954 | 1.323 | 4.588 | 1.462 | 3.101 | 1.238 | 3.970 | 1.338 | **2.687** | **1.147** | 3.041 | 1.186 |
| | 36 | 4.771 | 1.427 | 5.322 | 1.659 | 4.777 | 1.496 | 4.167 | 1.360 | 4.845 | 1.496 | 3.397 | 1.270 | 4.377 | 1.410 | **2.887** | **1.160** | 3.406 | 1.232 |
| | 48 | 4.945 | 1.462 | 5.425 | 1.632 | 5.333 | 1.592 | 4.746 | 1.463 | 4.865 | 1.516 | 2.947 | 1.203 | 4.811 | 1.503 | **2.797** | **1.155** | 3.459 | 1.221 |
| | 60 | 5.176 | 1.504 | 5.477 | 1.675 | 5.070 | 1.552 | 5.219 | 1.553 | 5.212 | 1.576 | 3.019 | 1.202 | 5.204 | 1.588 | **2.809** | **1.163** | 3.640 | 1.305 |
| Traffic | 24 | 0.668 | 0.378 | 0.648 | 0.403 | 0.506 | 0.278 | 0.597 | 0.332 | 0.608 | 0.334 | 0.550 | 0.363 | 0.606 | 0.338 | 0.562 | 0.375 | **0.491** | **0.274** |
| | 48 | 0.709 | 0.400 | 0.709 | 0.425 | **0.512** | **0.298** | 0.658 | 0.369 | 0.644 | 0.359 | 0.595 | 0.376 | 0.619 | 0.346 | 0.567 | 0.374 | 0.519 | 0.295 |
| | 168 | 0.900 | 0.523 | 0.713 | 0.435 | 0.521 | 0.319 | 0.664 | 0.363 | 0.660 | 0.391 | 0.649 | 0.407 | 0.635 | 0.347 | 0.607 | 0.385 | **0.513** | **0.289** |
| | 336 | 1.067 | 0.599 | 0.741 | 0.451 | 0.540 | 0.335 | 0.654 | 0.358 | 0.747 | 0.405 | 0.624 | 0.388 | 0.641 | 0.347 | 0.624 | 0.389 | **0.530** | **0.300** |
| | 720 | 1.461 | 0.787 | 0.768 | 0.474 | **0.557** | 0.343 | 0.685 | 0.370 | 0.792 | 0.430 | 0.674 | 0.417 | 0.670 | 0.364 | 0.623 | 0.378 | 0.573 | **0.313** |

$l$ are rearranged to get the layer prediction $\mathbf{x}^{pred,l}_{T+1:T+\tau}$. Predictions of all the layers are summed to obtain the final forecasting $\mathbf{x}^{pred}_{T+1:T+\tau}$.

## 4 EXPERIMENTS

### 4.1 PROTOCOLS

**Datasets** We conduct experiments on six real-world datasets following Zhou et al. (2021); Wu et al. (2021a). **1) ETTh1** (Electricity Transformer Temperature-hourly), **2) ETTm1** (Electricity Transformer Temperature-minutely), **3) WTH** (Weather), **4) ECL** (Electricity Consuming Load), **5) ILI** (Influenza-Like Illness), **6) Traffic**. The train/val/test splits for the first four datasets are same as Zhou et al. (2021), the last two are split by the ratio of 0.7:0.1:0.2 following Wu et al. (2021a).

**Baselines** We use the following popular models for MTS forecasting as baselines:**1) LSTMa** (Bahdanau et al., 2015), **2) LSTnet** (Lai et al., 2018), **3) MTGNN** (Wu et al., 2020), and recent Transformer-based models for MTS forecasting: **4) Transformer** (Vaswani et al., 2017), **5) Informer** (Zhou et al., 2021), **6) Autoformer** (Wu et al., 2021a), **7) Pyraformer** (Liu et al., 2021a) and **8) FEDformer** (Zhou et al., 2022).

**Setup** We use the same setting as in Zhou et al. (2021): train/val/test sets are zero-mean normalized with the mean and std of training set. On each dataset, we evaluate the performance over the changing future window size $\tau$. For each $\tau$, the past window size $T$ is regarded as a hyper-parameter to search which is a common protocol in recent MTS transformer literature (Zhou et al., 2021; Liu et al., 2021a). We roll the whole set with stride $= 1$ to generate different input-output pairs. The Mean Square Error (MSE) and Mean Absolute Error (MAE) are used as evaluation metrics. All experiments are repeated for 5 times and the mean of the metrics reported. Our Crossformer only utilize the past series to forecast the future, while baseline models use additional covariates such as hour-of-the-day. Details about datasets, baselines, implementation, hyper-parameters are shown in Appendix A.

### 4.2 MAIN RESULTS

As shown in Table 1, Crossformer shows leading performance on most datasets, as well as on different prediction length settings, with the 36 top-1 and 51 top-2 cases out of 58 in total. It is worth noting that, perhaps due to the explicit use of cross-dimension dependency via GNN, MTGNN outperforms many Transformer-based baselines. While MTGNN has been rarely compared in existing

Table 2: Component ablation of Crossformer: DSW embedding, TSA layer and HED on ETTh1.

| Models | Transformer | | DSW | | DSW+TSA | | DSW+HED | | DSW+TSA+HED | |
|---|---|---|---|---|---|---|---|---|---|---|
| Metric | MSE | MAE | MSE | MAE | MSE | MAE | MSE | MAE | MSE | MAE |
| 24 | 0.620 | 0.577 | 0.373 | 0.418 | 0.322 | 0.373 | 0.406 | 0.454 | **0.305** | **0.367** |
| 48 | 0.692 | 0.671 | 0.456 | 0.479 | 0.365 | 0.403 | 0.493 | 0.512 | **0.352** | **0.394** |
| 168 | 0.947 | 0.797 | 0.947 | 0.731 | 0.473 | 0.479 | 0.614 | 0.583 | **0.410** | **0.441** |
| 336 | 1.094 | 0.813 | 0.969 | 0.752 | 0.553 | 0.534 | 0.788 | 0.676 | **0.440** | **0.461** |
| 720 | 1.241 | 0.971 | 1.086 | 0.814 | 0.636 | 0.599 | 0.841 | 0.717 | **0.519** | **0.524** |

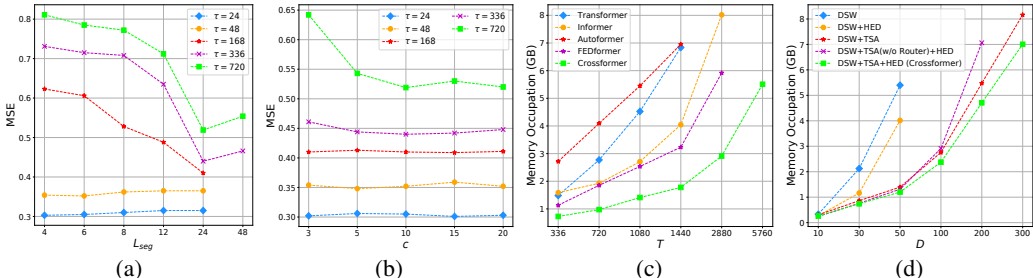

(a)      (b)      (c)      (d)

Figure 4: Evaluation on hyper-parameter impact and computational efficiency. (a) MSE against hyper-parameter segment length $L_{seg}$ in DSW embedding on ETTh1. (b) MSE against hyper-parameter number of routers $c$ in the Cross-Dimension Stage of TSA layer on ETTh1. (c) Memory occupation against the input length $T$ on ETTh1. (d) Memory occupation against number of dimensions $D$ on synthetic datasets with different number of dimensions.

transformers for MTS forecasting literatures. FEDformer and Autoformer outperform our model on ILI. We conjecture this is because the size of dataset ILI is small and these two models introduce the prior knowledge of sequence decomposition into the network structure which makes them perform well when the data is limited. Crossformer still outperforms other baselines on this dataset.

## 4.3 ABLATION STUDY

In our approach, there are three components: DSW embedding, TSA layer and HED. We perform ablation study on the ETTh1 dataset in line with Zhou et al. (2021); Liu et al. (2021a). We use Transformer as the baseline and **DSW+TSA+HED** to denote Crossformer without ablation. Three ablation versions are compared: **1) DSW 2) DSW+TSA 3) DSW+HED**.

We analyze the results shown in Table 2. **1)** DSW performs better than Transformer on most settings. The only difference between DSW and Transformer is the embedding method, which indicates the usefulness of DSW embedding and the importance of cross-dimension dependency. **2)** TSA constantly improves the forecasting accuracy. This suggests that it is reasonable to treat time and dimension differently. Moreover, TSA makes it possible to use Crossformer on datasets where the number of dimensions is large (e.g. $D = 862$ for dataset Traffic). **3)** Comparing DSW+HED with DSW, HED decreases the forecasting accuracy when prediction length is short but increases it for long term prediction. The possible reason is that information at different scales is helpful to long term prediction. **4)** Combining DSW, TSA and HED, our Crossformer yields best results on all settings.

## 4.4 EFFECT OF HYPER-PARAMETERS

We evaluate the effect of two hyper-parameters: segment length ($L_{seg}$ in Eq. 1) and number of routers in TSA ($c$ in Cross-Dimension Stage of TSA) on the ETTh1 dataset. **Segment Length:** In Fig. 4(a), we prolong the segment length from 4 to 24 and evaluate MSE with different prediction windows. For short-term forecasting ($\tau = 24, 48$), smaller segment yields relevantly better results, but the prediction accuracy is stable. For long-term forecasting ($\tau \geq 168$), prolonging the segment length from 4 to 24 causes the MSE to decrease. This indicates that long segments should be used for long-term forecasting. We further prolong the segment length to 48 for $\tau = 336, 720$, the MSE is slightly larger than that of 24. The possible reason is that 24 hours exactly matches the daily period of this dataset, while 48 is too coarse to capture fine-grained information. **Number of Routers in TSA Layer:** Number of Routers $c$ controls the information bandwidth among all dimensions. As Fig. 4(b) shows, the performance of Crossformer is stable w.r.t to $c$ for $\tau \leq 336$. For $\tau = 720$, the MSE is

large when $c = 3$ but decreases and stabilizes when $c \geq 5$. In pratice, we set $c = 10$ to balance the prediction accuracy and computation efficiency.

## 4.5 COMPUTATIONAL EFFICIENCY ANALYSIS

The theoretical complexity per layer of Transformer-based models is compared in Table 3. The complexity of Crossformer encoder is quadratic w.r.t $T$. However, for long-term prediction where large $L_{seq}$ is used, the coefficient $\frac{1}{L_{seq}^2}$ term can significantly reduce its practical complexity. We evaluate the memory occupation of these models on ETTh1.[4] We set the prediction window $\tau = 336$ and prolong input length $T$. For Crossformer, $L_{seg}$ is set to 24, which is the best value for $\tau \geq 168$

Table 3: Computation complexity per layer of Transformer-based models. $T$ denotes the length of past series, $\tau$ denotes the length of prediction window, $D$ denotes the number of dimensions, $L_{seg}$ denotes the segment length of DSW embedding in Crossformer.

| Method | Encoder layer | Decoder layer |
|---|---|---|
| Transformer (Vaswani et al., 2017) | $O(T^2)$ | $O(\tau(\tau + T))$ |
| Informer (Zhou et al., 2021) | $O(T \log T)$ | $O(\tau(\tau + \log T))$ |
| Autoformer (Wu et al., 2021a) | $O(T \log T)$ | $O((\frac{T}{2} + \tau) \log (\frac{T}{2} + \tau))$ |
| Pyraformer (Liu et al., 2021a) | $O(T)$ | $O(\tau(\tau + T))$ |
| FEDformer (Zhou et al., 2022) | $O(T)$ | $O(\frac{T}{2} + \tau)$ |
| Crossformer (Ours) | $O\left(\frac{D}{L_{seg}^2} T^2\right)$ | $O\left(\frac{D}{L_{seg}^2} \tau(\tau + T)\right)$ |

(see Fig. 4 (a)). The result in Fig. 4 (c) shows that Crossformer achieves the best efficiency among the five methods within the tested length range. Theoretically, Informer, Autoformer and FEDformer are more efficient when $T$ approaches infinity. In practice, Crossformer performs better when $T$ is not extremely large (e.g. $T \leq 10^4$).

We also evaluate the memory occupation w.r.t the number of dimensions $D$. For baseline models where cross-dimension dependency is not modeled explicitly, $D$ has little effect. Therefore, we compare Crossformer with its ablation versions in Section 4.3. We also evaluate the TSA layers that directly use MSA in Cross-Dimension Stage without the Router mechanism, denoted as TSA(w/o Router). Fig. 4 (d) shows that Crossformer without TSA layer (DSW and DSW+HED) has quadratic complexity w.r.t $D$. TSA(w/o Router) helps to reduce complexity and the Router mechanism further makes the complexity linear, so that Crossformer can process data with $D = 300$. Moreover, HED can slightly reduce the memory cost and we analyze this is because there are less vectors in upper layers after segment merging (see Fig. 3). Besides memory occupation, the actual running time evaluation is shown in Appendix B.6.

## 5 CONCLUSIONS AND FUTURE WORK

We have proposed Crossformer, a Transformer-based model utilizing cross-dimension dependency for multivariate time-series (MTS) forecasting. Specifically, the Dimension-Segment-Wise (DSW) embedding embeds the input data into a 2D vector array to preserve the information of both time and dimension. The Two-Stage-Attention (TSA) layer is devised to capture the cross-time and cross-dimension dependency of the embedded array. Using DSW embedding and TSA layer, a Hierarchical Encoder-Decoder (HED) is devised to utilize the information at different scales. Experimental results on six real-world datasets show its effectiveness over previous state-of-the-arts.

We analyzed the limitations of our work and briefly discuss some directions for future research: **1)** In Cross-Dimension Stage, we build a simple full connection among dimensions, which may introduce noise on high-dimensional datasets. Recent sparse and efficient Graph Transformers (Wu et al., 2022) can benefit our TSA layer on this problem. **2)** A concurrent work (Zeng et al., 2023) which was accepted after the submission of this work received our attention. It questions the effectiveness of Transformers for MTS forecasting and proposes DLinear that outperforms all Transformers including our Crossformer on three of the six datasets (details are in Appendix B.2). It argues the main reason is that MSA in Transformer is permutation-invariant. Therefore, enhancing the ordering preserving capability of Transformers is a promising direction to overcome this shortcoming . **3)** Considering datasets used in MTS analysis are much smaller and simpler than those used in vision and texts, besides new models, large datasets with various patterns are also needed for future research.

---

[4]Pyraformer is not evaluated as it requires the additional compiler TVM to achieve linear complexity.

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

# A DETAILS OF EXPERIMENTS

## A.1 BENCHMARKING DATASETS

We conduct experiments on the following six real-world datasets following Zhou et al. (2021); Wu et al. (2021a):

**1) ETTh1** (Electricity Transformer Temperature-hourly) contains 7 indicators of an electricity transformer in two years, including oil temperature, useful load, etc. Data points are recorded every hour and train/val/test is 12/4/4 months.

**2) ETTm1** (Electricity Transformer Temperature-minutely) contains the same indicators as ETTh1 but data points are recorded every 15 miniutes. Train/val/test split is same as ETTh1.

**3) WTH** (Weather) contains 12 meteorological indicators in U.S. in 4 years, including visibility, wind speed, etc. Train/val/test is 28/10/10 months.

**4) ECL** (Electricity Consuming Load) contains hourly electricity consumption (in Kwh) of 321 clients in two years. Train/val/test is 15/3/4 months.

**5) ILI** (Influenza-Like Illness) contains 7 weekly recorded indicators of patients data from Centers for Disease Control and Prevention of the United States between between 2002 and 2021. The ratio of train/validation/test splits is 0.7:0.1:0.2.

**6) Traffic** contains hourly road occupancy rates measured by 862 sensors on San Francisco Bay area freeways in 2 years. The ratio of train/validation/test splits is 0.7:0.1:0.2.

The train/val/test splits for ETTh1, ETTm1, WTH, ECL are same as Zhou et al. (2021), for ILI and Traffic are same as Wu et al. (2021a).

The first four datasets are publicly available at https://github.com/zhouhaoyi/Informer2020 and the last two are publicly available at https://github.com/thuml/Autoformer.

## A.2 BASELINE METHODS

We briefly describe the selected baselines:

**1) LSTMa** (Bahdanau et al., 2015) treats the input MTS as a sequence of multi-dimensional vectors. It builds an encoder-decoder using RNN and automatically aligns target future steps with their relevant past.

**2) LSTnet** (Lai et al., 2018) uses CNN to extract cross-dimension dependency and short term cross-time dependency. The long-term cross-time dependency is captured through RNN. The source code is available at https://github.com/laiguokun/LSTNet.

**3) MTGNN** (Wu et al., 2020) explicitly utilizes cross-dimension dependency using GNN. A graph learning layer learns a graph structure where each node represents one dimension in MTS. Then graph convolution modules are interleaved with temporal convolution modules to explicitly capture cross-dimension and cross-time dependency respectively. The source code is available at https://github.com/nnzhan/MTGNN.

**4) Transformer** is closed to the original Transformer (Vaswani et al., 2017) that uses self-attention mechanism to capture cross-time dependency. The Informer-style one-step generative decoder is used for forecasting, therefore this is denoted as Informer[†] in Informer (Zhou et al., 2021).

**5) Informer** (Zhou et al., 2021) is a Transformer-based model using the ProbSparse self-attention to capture cross-time dependency for forecasting. The source code of Transformer and Informer is available at https://github.com/zhouhaoyi/Informer2020.

**6) Autoformer** (Wu et al., 2021a) is a Transformer-based model using decomposition architecture with Auto-Correlation mechanism to capture cross-time dependency for forecasting. The source code is available at https://github.com/thuml/Autoformer.

**7) Pyraformer** (Liu et al., 2021a) is a Transformer-based model learning multi-resolution representation of the time series by the pyramidal attention module to capture cross-time dependency for forecasting. The source code is available at https://github.com/alipay/Pyraformer.

**8) FEDformer** (Zhou et al., 2022) is a Transformer-based model that uses the seasonal-trend decomposition with frequency enhanced blocks to capture cross-time dependency for forecasting. The source code is available at https://github.com/MAZiqing/FEDformer.

### A.3 Hyper-parameter Selection and Implementation Details

#### A.3.1 Main Experiments

For the main experiments, we use the Crossformer with 3 encoder layers. The number of routers in TSA layer $c$ is set to 10. For dataset ETTh1, ETTm1, WTH and ILI, dimension of hidden state $d_{model}$ is set to 256, the head number of multi-head attention is set to 4; For dataset ECL and Traffic, dimension of hidden state $d_{model}$ is set to 64, the head number of multi-head attention is set to 2. The segment length $L_{seg}$ is chosen from $\{6, 12, 24\}$ via grid search. We use MSE as loss function and batch size is set to 32. Adam optimizer is used for training and the initial learning rate is chosen from {5e-3, 1e-3, 5e-4, 1e-4, 5e-5, 1e-5} via grid search. The total number of epochs is 20. If the validation loss does not decreases within three epochs, the training process will stop early.

For baseline models, if the original papers conduct experiments on the dataset we use, the hyper-parameters (except input length $T$) recommended in the original papers are used, including the number of layers, dimension of hidden states, etc. Otherwise, the hyper-parameters are chosen through grid search using the validation set.

Following Zhou et al. (2021), on datasets ETTh1, WTH, ECL and Traffic, for different prediction length $\tau$, the input length $T$ is chosen from $\{24, 48, 96, 168, 336, 720\}$; on ETTm1, the input length is chosen from $\{24, 48, 96, 192, 288, 672\}$; on ILI, the input length is chosen from $\{24, 36, 48, 60\}$.

All models including Crossformer and baselines are implemented in PyTorch and trained on a single NVIDIA Quadro RTX 8000 GPU with 48GB memory.

#### A.3.2 Efficiency Analysis

To evaluate the computational efficiency w.r.t the input length $T$ in Figure 4(c) of the main paper, we align the hyper-parameters of all Transformer-based models as follows: prediction length $\tau$ is set to 336, number of encoder layers is set to 2, dimension of hidden state $d_{model}$ is set to 256, the head number of multi-head attention is set to 4.

To evaluate the computational efficiency w.r.t the number of dimensions $D$ in Figure 4(d) of the main paper, we align the hyper-parameters of ablation versions of Crossformer as follows as: both input length $T$ and prediction length $\tau$ are set to 336, number of encoder layers is set to 3, $d_{model}$ is set to 64, the head number of multi-head attention is set to 2.

Experiments in the computational efficiency analysis section are conducted on a single NVIDIA GeForce RTX 2080Ti GPU with 11GB memory.

### A.4 Details of Ablation Versions of Crossformer

We describe the models we used in ablation study below:

**1) DSW** represents Crossformer without TSA and HED. The input is embedded by DSW embedding and flatten into a 1D sequence to be input to the original Transformer. The only difference between this model and the Transformer is the embedding method.

**2) DSW+TSA** represents Crossformer without HED. Compared with Crossformer, the encoder does not use segment merging to capture dependency at different scales. The decoder takes the final output of encoder (i.e. $\mathbf{Z}^{enc,N}$) as input instead of using encoder's output at each scale.

**3) DSW+HED** represents Crossformer without TSA. In each encoder layer and decoder layer, the 2D vector array is flatten into a 1D sequence to be input to the original self-attention layer for dependency capture.

## B Extra Experimental Results

### B.1 Showcases of Main Results

Figure 5 shows the forecasting cases of three dimensions of the ETTm1 dataset with prediction length $\tau = 288$. For dimension "HUFL", all the five models capture the periodic pattern, but Crossformer is

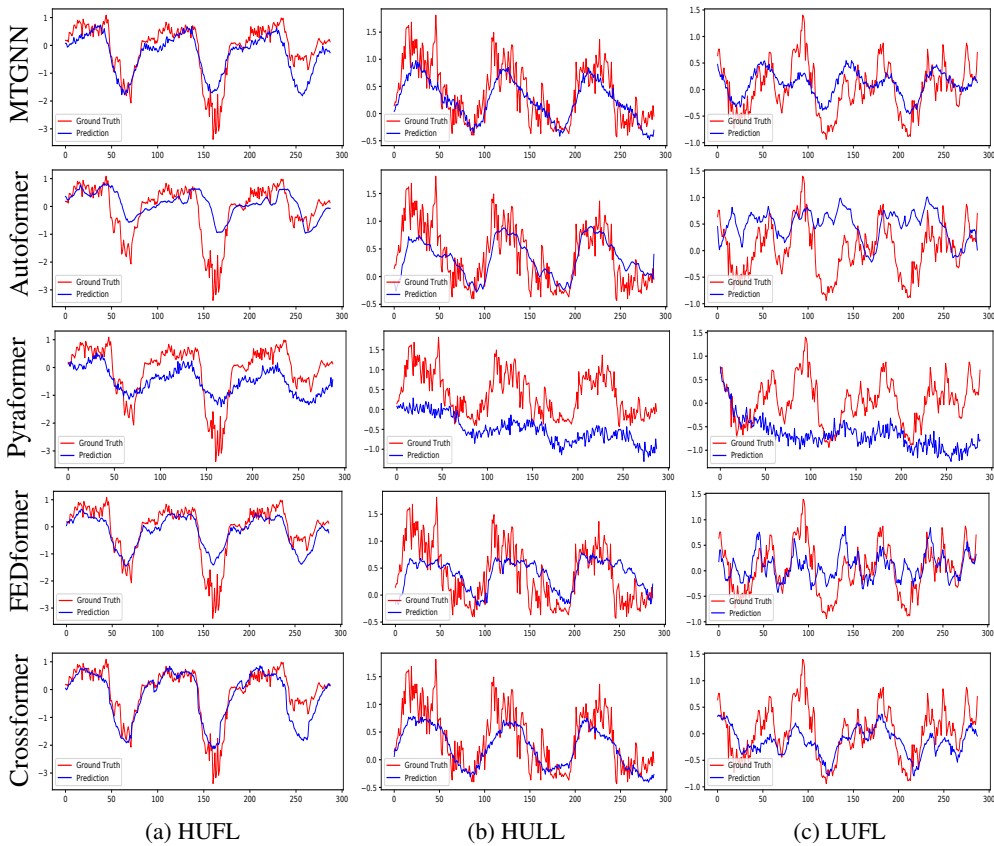

Figure 5: Forecasting cases of three dimensions: High UseFul Load (HUFL), High UseLess Load (HULL) and Low UseFul Load (LUFL) of the ETTm1 dataset with prediction length $\tau = 288$. The red / blue curves stand for the ground truth / prediction. Each row represents one model and each column represents one dimension.

the closest to the ground truth. For "HULL", Pyraformer fails to capture the periodic pattern from the noisy data. For "LUFL" where the data has no clear periodic pattern, MTGNN, FEDformer and Crossformer capture its trend and show significantly better results than the other two models.

Figure 6 shows the forecasting cases of three dimensions of the WTH dataset with prediction length $\tau = 336$. For dimension "DBT", all the five models capture the periodic pattern. For "DPT", Autoformer and FEDformer fails to capture increasing trend of the data. For "WD", all models capture the periodic pattern from the noisy data, and the cruves output by MTGNN and Crossformer are sharper than the other three models.

### B.2  COMPARISON WITH EXTRA METHODS

We further compare with two additional concurrent methods which were either not peer-reviewed (Grigsby et al., 2022) or were accepted after the submission of this work (Zeng et al., 2023): **1) STformer** (Grigsby et al., 2022), a Transformer-based model that directly flattens the multivariate time-series $\mathbf{x}_{1:T} \in \mathbb{R}^{T \times D}$ into a 1D sequence to be input to Transformers; **2) DLinear** (Zeng et al., 2023), a simple linear model with seasonal-trend decomposition that challenges Transformer-based models for MTS forecasting. Results are shown in Table 4 and LSTMa and LSTnet are omitted as they are not competitive with other models.

The basic idea of STformer is similar to our Crossformer: both of them extend the 1-D attention to 2-D. The explicit utilization of cross-dimension dependency makes STformer competitive with previous Transformer-based models on ETTh1, ETTm1 and WTH, especially for short-term prediction. However, STformer directly flattens the raw 2-D time series into a 1-D sequence to be input to the

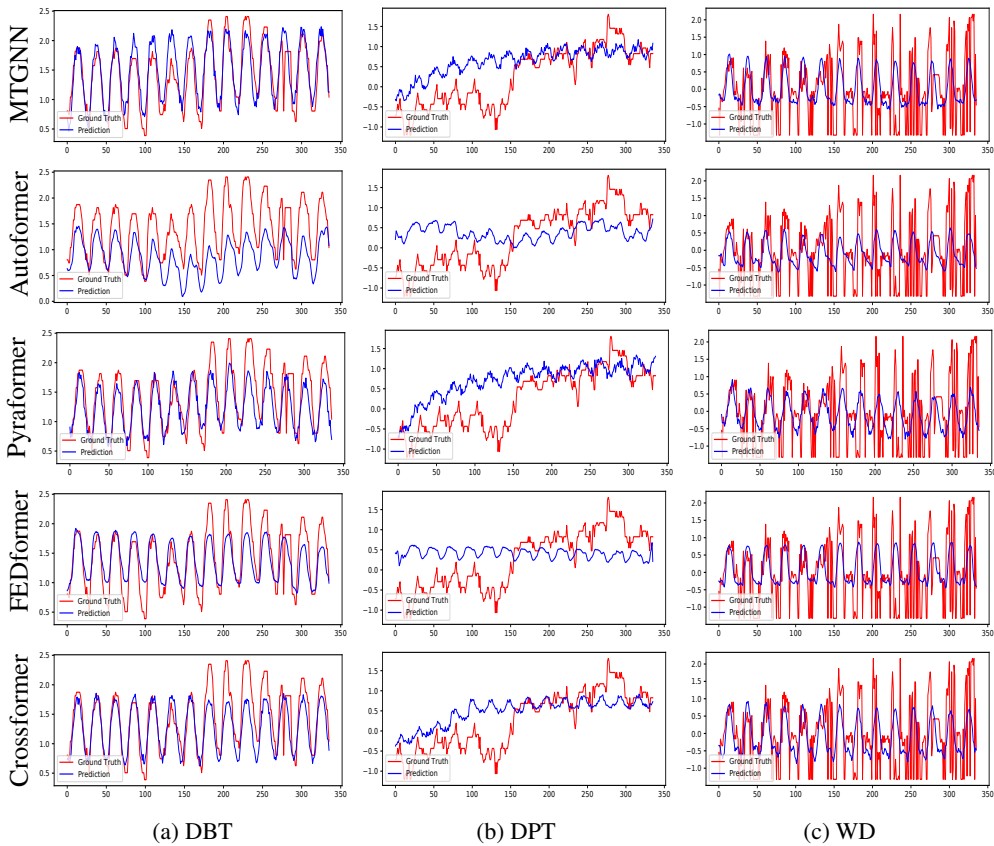

Figure 6: Forecasting cases of three dimensions: Dry Bulb Temperature (DBT), Dew Point Temperature (DPT) and Wind Direction (WD) of the WTH dataset with prediction length $\tau = 336$. The red / blue curves stand for the ground truth / prediction. Each row represents one model and each column represents one dimension.

Transformer. This straightforward method does not distinguish the time and dimension axes and is computationally inefficient. Therefore, besides the good performance for short-term prediction, STformer has difficulty in long-term prediction and encounters the out-of-memory (OOM) problem on high-dimensional datasets (ECL and Traffic). While Crossformer uses the DSW embedding to capture local dependency and reduce the complexity. The TSA layer with the router mechanism is devised to deal with the heterogeneity of time and dimension axis and further improve efficiency.

DLinear is on par with our Crossformer on ETTh1 and ETTm1 ($\tau \leq 96$); has similar performance with FEDformer on ILI; performs worse than Crossformer on WTH; outperforms all Transformer-based models including our Crossformer on ETTm1 ($\tau \geq 288$), ECL and Traffic. Considering its simplicity, the performance is impressive. Based on the results, we analyze the limitations of Crossformer and propose some directions to improve it in the future:

**1)** In Cross-Dimension Stage of TSA layer, we simply build an all-to-all connection among $D$ dimensions with the router mechanism. Besides capturing the cross-dimension dependency, this full connection also introduces noise, especially for high-dimensional dataset. We think high-dimensional data has the sparse property: each dimension is only relevant to a small fraction of all dimensions. Therefore, utilizing the sparsity to reduce noise and improve the computation efficiency of the TSA layer could be a promising direction.

**2)** Authors of DLinear (Zeng et al., 2023) argue that the Transformer-based models have difficulty in preserving ordering information because the attention mechanism is permutation-invariant and the absolute position embedding injected into the model is not enough for time series forecasting, which is an order-sensitive task. Although Yun et al. (2020) theoretically proves that Transformers with

Table 4: MSE/MAE comparison with extra methods: STformer (Grigsby et al., 2022) and DLinear (Zeng et al., 2023). Bold/underline indicates the best/second. OOM indicates out-of-memory problem. Gray background marks the CNN-GNN-based model; yellow marks Transformer-based models where cross-dimension dependency is omitted; blue marks Transformer-based models explicitly utilizing cross-dimension dependency; red marks the linear model with series decomposition.

| Models | MTGNN | | Transformer | | Informer | | Autoformer | | Pyraformer | | FEDformer | | STformer | | Crossformer | | DLinear | |
|---|---|---|---|---|---|---|---|---|---|---|---|---|---|---|---|---|---|---|
| Metric | MSE | MAE | MSE | MAE | MSE | MAE | MSE | MAE | MSE | MAE | MSE | MAE | MSE | MAE | MSE | MAE | MSE | MAE |
| **ETTh1** 24 | 0.336 | 0.393 | 0.620 | 0.577 | 0.577 | 0.549 | 0.439 | 0.440 | 0.493 | 0.507 | 0.318 | 0.384 | 0.368 | 0.441 | **0.305** | 0.367 | 0.312 | **0.355** |
| 48 | 0.386 | 0.429 | 0.692 | 0.671 | 0.685 | 0.625 | 0.429 | 0.442 | 0.554 | 0.544 | **0.342** | 0.396 | 0.445 | 0.465 | 0.352 | 0.394 | 0.352 | **0.383** |
| 168 | 0.466 | 0.474 | 0.947 | 0.797 | 0.931 | 0.752 | 0.493 | 0.479 | 0.781 | 0.675 | 0.412 | 0.449 | 0.652 | 0.608 | **0.410** | 0.441 | 0.416 | **0.430** |
| 336 | 0.736 | 0.643 | 1.094 | 0.813 | 1.128 | 0.873 | 0.509 | 0.492 | 0.912 | 0.747 | 0.456 | 0.474 | 1.069 | 0.806 | **0.440** | 0.461 | 0.450 | **0.452** |
| 720 | 0.916 | 0.750 | 1.241 | 0.917 | 1.215 | 0.896 | 0.539 | 0.537 | 0.993 | 0.792 | 0.521 | 0.515 | 1.071 | 0.817 | 0.519 | 0.524 | **0.486** | **0.501** |
| **ETTm1** 24 | 0.260 | 0.324 | 0.306 | 0.371 | 0.323 | 0.369 | 0.410 | 0.428 | 0.310 | 0.371 | 0.290 | 0.364 | 0.278 | 0.348 | **0.211** | 0.293 | 0.217 | 0.289 |
| 48 | 0.386 | 0.408 | 0.465 | 0.470 | 0.494 | 0.503 | 0.485 | 0.464 | 0.465 | 0.464 | 0.342 | 0.396 | 0.445 | 0.458 | 0.300 | 0.352 | **0.278** | **0.330** |
| 96 | 0.428 | 0.446 | 0.681 | 0.612 | 0.678 | 0.614 | 0.502 | 0.476 | 0.520 | 0.504 | 0.366 | 0.412 | 0.420 | 0.455 | 0.320 | 0.373 | **0.310** | **0.354** |
| 288 | 0.469 | 0.488 | 1.162 | 0.879 | 1.056 | 0.786 | 0.604 | 0.522 | 0.729 | 0.657 | 0.398 | 0.433 | 0.733 | 0.597 | 0.404 | 0.427 | **0.369** | **0.386** |
| 672 | 0.620 | 0.571 | 1.231 | 1.103 | 1.192 | 0.926 | 0.607 | 0.530 | 0.980 | 0.678 | 0.455 | 0.464 | 0.777 | 0.625 | 0.569 | 0.528 | **0.416** | **0.417** |
| **WTH** 24 | 0.307 | 0.356 | 0.349 | 0.397 | 0.335 | 0.381 | 0.363 | 0.396 | 0.301 | 0.359 | 0.357 | 0.412 | 0.307 | 0.359 | **0.294** | **0.343** | 0.357 | 0.391 |
| 48 | 0.388 | 0.422 | 0.386 | 0.433 | 0.395 | 0.459 | 0.456 | 0.462 | 0.376 | 0.421 | 0.428 | 0.458 | 0.381 | 0.416 | **0.370** | **0.411** | 0.425 | 0.444 |
| 168 | 0.498 | 0.512 | 0.613 | 0.582 | 0.608 | 0.567 | 0.574 | 0.548 | 0.519 | 0.521 | 0.564 | 0.541 | 0.497 | 0.502 | **0.473** | **0.494** | 0.515 | 0.516 |
| 336 | 0.506 | 0.523 | 0.707 | 0.634 | 0.702 | 0.620 | 0.600 | 0.571 | 0.539 | 0.543 | 0.533 | 0.536 | 0.566 | 0.564 | **0.495** | **0.515** | 0.536 | 0.537 |
| 720 | **0.510** | **0.527** | 0.834 | 0.741 | 0.831 | 0.731 | 0.587 | 0.570 | 0.547 | 0.553 | 0.562 | 0.557 | 0.589 | 0.582 | 0.526 | 0.542 | 0.582 | 0.571 |
| **ECL** 48 | 0.173 | 0.280 | 0.334 | 0.399 | 0.344 | 0.393 | 0.241 | 0.351 | 0.478 | 0.471 | 0.229 | 0.338 | 0.356 | 0.432 | 0.156 | **0.255** | **0.155** | 0.258 |
| 168 | 0.236 | 0.320 | 0.353 | 0.420 | 0.368 | 0.424 | 0.299 | 0.387 | 0.452 | 0.455 | 0.263 | 0.361 | 0.516 | 0.527 | 0.231 | 0.309 | **0.195** | **0.287** |
| 336 | 0.328 | 0.373 | 0.381 | 0.439 | 0.381 | 0.431 | 0.375 | 0.428 | 0.463 | 0.456 | 0.305 | 0.386 | OOM | | 0.323 | 0.369 | **0.238** | **0.316** |
| 720 | 0.422 | 0.410 | 0.391 | 0.438 | 0.406 | 0.443 | 0.377 | 0.434 | 0.480 | 0.461 | 0.372 | 0.434 | OOM | | 0.404 | 0.423 | **0.272** | **0.346** |
| 960 | 0.471 | 0.451 | 0.492 | 0.550 | 0.460 | 0.548 | 0.366 | 0.426 | 0.550 | 0.489 | 0.393 | 0.449 | OOM | | 0.433 | 0.438 | **0.299** | **0.367** |
| **ILI** 24 | 4.265 | 1.387 | 3.954 | 1.323 | 4.588 | 1.462 | 3.101 | 1.238 | 3.970 | 1.338 | **2.687** | **1.147** | 3.150 | 1.232 | 3.041 | 1.186 | 2.940 | 1.205 |
| 36 | 4.777 | 1.496 | 4.167 | 1.360 | 4.845 | 1.496 | 3.397 | 1.270 | 4.377 | 1.410 | **2.887** | **1.160** | 3.512 | 1.243 | 3.406 | 1.232 | 2.826 | 1.184 |
| 48 | 5.333 | 1.592 | 4.746 | 1.463 | 4.865 | 1.516 | 2.947 | 1.203 | 4.811 | 1.503 | 2.797 | **1.155** | 3.499 | 1.234 | 3.459 | 1.221 | 2.677 | 1.155 |
| 60 | 5.070 | 1.552 | 5.219 | 1.553 | 5.212 | 1.576 | 3.019 | 1.202 | 5.204 | 1.588 | **2.809** | **1.163** | 3.715 | 1.316 | 3.640 | 1.305 | 3.011 | 1.245 |
| **Traffic** 24 | 0.506 | 0.278 | 0.597 | 0.332 | 0.608 | 0.334 | 0.550 | 0.363 | 0.606 | 0.338 | 0.562 | 0.375 | 0.747 | 0.447 | 0.491 | 0.274 | **0.351** | **0.261** |
| 48 | 0.512 | 0.298 | 0.658 | 0.369 | 0.644 | 0.359 | 0.595 | 0.376 | 0.619 | 0.346 | 0.567 | 0.374 | OOM | | 0.519 | 0.295 | **0.370** | **0.270** |
| 168 | 0.521 | 0.319 | 0.664 | 0.363 | 0.660 | 0.391 | 0.649 | 0.407 | 0.635 | 0.347 | 0.607 | 0.385 | OOM | | 0.513 | 0.289 | **0.395** | **0.277** |
| 336 | 0.540 | 0.335 | 0.654 | 0.358 | 0.747 | 0.405 | 0.624 | 0.388 | 0.641 | 0.347 | 0.624 | 0.389 | OOM | | 0.530 | 0.300 | **0.415** | **0.289** |
| 720 | 0.557 | 0.343 | 0.685 | 0.370 | 0.792 | 0.430 | 0.674 | 0.417 | 0.670 | 0.364 | 0.623 | 0.378 | OOM | | 0.573 | 0.313 | **0.455** | **0.313** |

trainable positional embedding are universal approximators of sequence-to-sequence functions, the ordering information still needs to be enhanced in practice. We think that relative position encoding in texts (Ke et al., 2021; Dufter et al., 2022) and vision (Wu et al., 2021b) could be useful for ordering information enhancement.

**3)** The sizes of datasets used for time series forecasting are much smaller than those for texts and vision, and the patterns in time series datasets are also simpler. Considering vision transformers surpass inductive bias and achieves excellent results compared to CNNs after pre-trained on large amounts of data (Dosovitskiy et al., 2021), Transformers for time series may also require large-size datasets with various patterns to exploit their full potential.

As quoted from the paper, authors mentioned that DLinear "does not model correlations among variates". Therefore, incorporating cross-dimension dependency into DLinear to further improve prediction accuracy is also a promising direction. Moreover, our DSW embedding to enhance locality and HED to capture dependency at different scales can also be potentially useful to further inspire and enhance DLinear.

## B.3 ABLATION STUDY OF THE ROUTER MECHANISM

The ablation study of the three main components of Crossformer is shown in Sec. 4.3. In this section, we conduct an ablation study of the router mechanism, a sub-module in TSA layer, and evaluate its impact on prediction accuracy. It should be noticed that the router mechanism is mainly proposed to reduce the computation complexity when $D$ is large. Results are shown in Table 5. Adding TSA(w/o Router) constantly improves the prediction accuracy of DSW and DSW+HED, showing the necessity of capturing cross-time and cross-dimension dependency in two different stages. For short term prediction ($\tau \leq 168$), the performances of TSA(w/o Router) and TSA are similar, no matter whether HED is used or not. For long term prediction ($\tau \geq 336$), the router mechanism slightly improves the prediction accuracy. The possible reason is that we set separate routers for each time step, which helps capture long-term dependency that varies over time.

Table 5: Complementary results to ablation study in Table 2. TSA(w/o Router) denotes TSA layer without the router mechanism that directly uses MSA in the Cross-Dimension Stage.

| Models | DSW | | DSW+ TSA(w/o Router) | | DSW+TSA | | DSW+HED | | DSW+HED+ TSA(w/o Router) | | DSW+TSA+HED | |
|---|---|---|---|---|---|---|---|---|---|---|---|---|
| Metric | MSE | MAE | MSE | MAE | MSE | MAE | MSE | MAE | MSE | MAE | MSE | MAE |
| 24 | 0.373 | 0.418 | 0.320 | 0.376 | 0.322 | 0.373 | 0.406 | 0.454 | 0.311 | 0.375 | **0.305** | **0.367** |
| 48 | 0.456 | 0.479 | 0.356 | 0.396 | 0.365 | 0.403 | 0.493 | 0.512 | 0.363 | 0.406 | **0.352** | **0.394** |
| 168 | 0.947 | 0.731 | 0.487 | 0.493 | 0.473 | 0.479 | 0.614 | 0.583 | 0.416 | 0.444 | **0.410** | **0.441** |
| 336 | 0.969 | 0.752 | 0.585 | 0.564 | 0.553 | 0.534 | 0.788 | 0.676 | 0.487 | 0.499 | **0.440** | **0.461** |
| 720 | 1.086 | 0.814 | 0.665 | 0.615 | 0.636 | 0.599 | 0.841 | 0.717 | 0.540 | 0.542 | **0.519** | **0.524** |

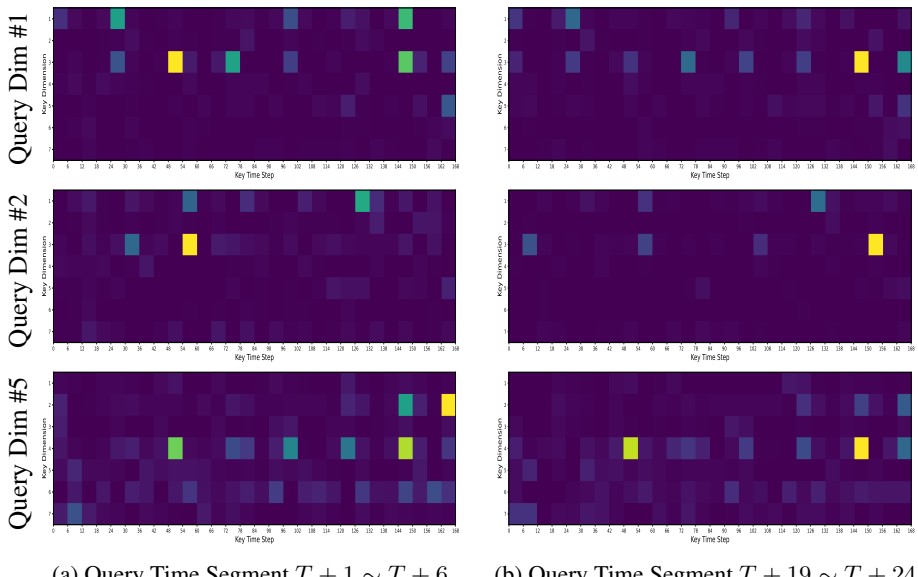

(a) Query Time Segment $T + 1 \sim T + 6$        (b) Query Time Segment $T + 19 \sim T + 24$

Figure 7: Attention scores calculated by the decoder of the ablation version of Crossformer (i.e. DSW) on dataset ETTh1. The input length, prediction length and segment length are set as $T = 168, \tau = 24, L_{seg} = 6$. The x axis in each sub-figure represents the time steps serve as keys in attention mechanism, while the y axis denotes dimensions. Brighter color denotes higher attention weights.

## B.4 DEPENDENCY VISUALIZATION

As the attention scores computed by Crossformer are abstract and hard to visualize, we visualize scores computed by the ablation version, DSW, in Figure 7. In addition to cross-time dependency that other Transformer models can compute, Crossformer also provides information about cross-dimension dependency. As shown in Figure 7, when predicting Dim #1, the model focus on both Dim #1 and #3. When predicting Dim #5, instead of focus on Dim #5 itself, more attention is paid to Dim #4.

## B.5 HIERARCHICAL PREDICTION PATTERN VISUALIZATION

Figure 8 shows the hierarchical prediction patterns output by our HED. The top prediction layer, Layer 3, captures the low frequency general trend and periodic pattern of the future value. By adding predictions at finer scales, finer high frequency patterns are added and the prediction get closer to the ground truth curve.

## B.6 RUNNING TIME EFFICIENCY ANALYSIS

In the main paper, we show the memory occupation w.r.t input length $T$ and number of dimensions $D$. Here we evaluate the running time. Figure 9 (a) shows the running time per batch of Crossformer

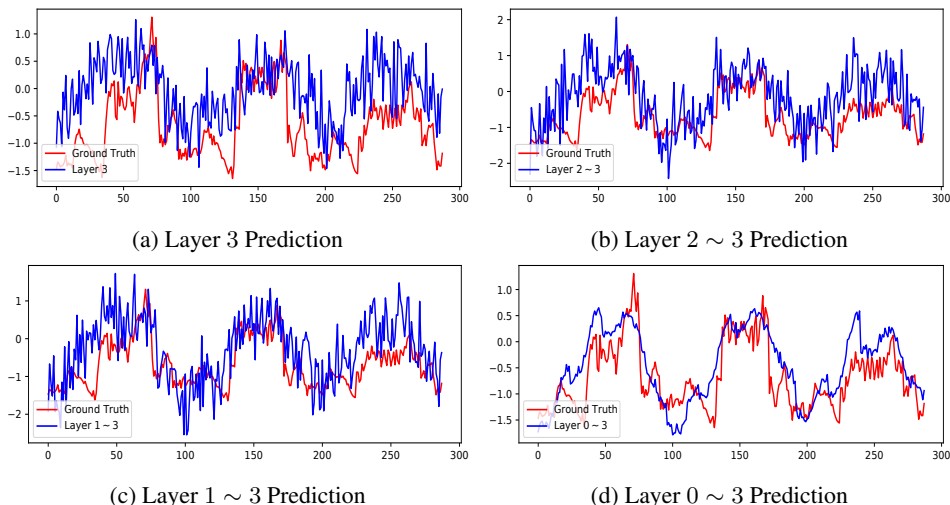

(a) Layer 3 Prediction

(b) Layer $2 \sim 3$ Prediction

(c) Layer $1 \sim 3$ Prediction

(d) Layer $0 \sim 3$ Prediction

Figure 8: Hierarchical prediction visualization of ETTm1 with dimension HUFL and prediction length $\tau = 288$. From top left to bottom right, we gradually add layer predictions at finer scales.

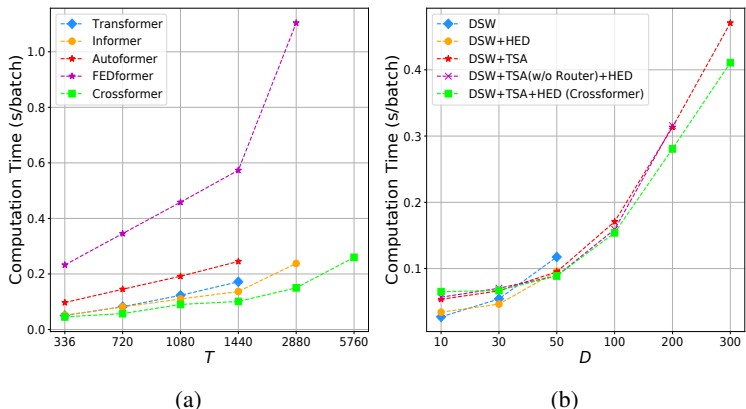

(a)

(b)

Figure 9: Evaluation on computational speed. (a) Running time per batch w.r.t the input length $T$ on ETTh1. (b) Running time per batch w.r.t number of dimensions $D$ on synthetic datasets by different numbers of dimensions.

and other Transformer-based models w.r.t input length $T$. FEDformer is much slower than other Transformer-based models. Crossformer achieves the best computation speed among the five methods within the tested length range.

Figure 9 (b) shows the running time per batch of Crossformer and its ablation versions w.r.t the number of dimensions $D$. Crossformers without TSA layer (DSW and DSW+HED) are faster when $D$ is small ($D \leq 30$). However, they have difficulty processing high-dimensional MTS due to the quadratic complexity w.r.t $D$. Indeed, for a single NVIDIA GeForce RTX 2080Ti GPU with 11GB memory, DSW and DSW+HED encounters the out-of-memory (OOM) problem when $D > 50$. Moreover, TSA(w/o Router) encounter the OOM problem when $D > 200$.

## C  DISCUSSION ON THE SELECTION OF HYPER-PARAMETERS

We recommend to first determine the segment length $L_{seg}$, as it is related to both the model performance and computation efficiency. The general idea is to use small $L_{seg}$ for short-term prediction and large $L_{seg}$ for long-term prediction. Some priors about the data also help to select $L_{seg}$. For example, if the hourly sampled data has a daily period, it is better to set $L_{seg} = 24$. Next, we select the number

Table 6: MSE and MAE evaluation with different segment lengths on ETTm1 dataset. * denotes segment length used in the main text, which is a divisor of $T, \tau$.

| Metric | MSE | MAE | MSE | MAE | MSE | MAE |
|---|---|---|---|---|---|---|
| Segment Length $L_{seg}$ | 5 | | 6* | | 7 | |
| $T = 288, \tau = 48$ | 0.291 | 0.349 | 0.300 | 0.352 | 0.284 | 0.346 |
| Segment Length $L_{seg}$ | 22 | | 24* | | 26 | |
| $T = 672, \tau = 288$ | 0.401 | 0.424 | 0.404 | 0.427 | 0.409 | 0.429 |

of layers for encoder and decoder $N$. Crossformer with larger $N$ can utilize information of more scales, but also requires more computing resources. The number of routers in TSA layer $c$ can be set to 5 or 10 to balance the prediction accuracy and computation efficiency. Finally, dimension of hidden states $d_{model}$ and head number of multi-head attention can be determined based on the available computing resources.

## D  SUPPLEMENTARY DESIGN TO CROSSFORMER

### D.1  HANDLING INDIVISIBLE LENGTH

In the main paper, we assume that the input length $T$ and prediction length $\tau$ are divisible by segment length $L_{seg}$. In this section, we use padding mechanism to handle cases where the assumption is not satisfied.

If $T$ is not divisible by $L_{seg}$, we have $(k_1 - 1)L_{seg} < T < k_1 L_{seg}$ for some $k_1$. We pad $k_1 L_{seg} - T$ duplicated $\mathbf{x}_1$ in front of $\mathbf{x}_{1:T}$ to get $\mathbf{x}'_{1:T}$:

$$\mathbf{x}'_{1:T} = \big[ \underbrace{\mathbf{x}_1, \ldots, \mathbf{x}_1}_{k_1 L_{seg} - T}, \mathbf{x}_{1:T} \big] \tag{9}$$

where $[,]$ denotes the concatenation operation. $\mathbf{x}'_{1:T} \in \mathbb{R}^{k_1 L_{seg} \times D}$ can be input to the encoder of Crossformer.

If $\tau$ is not divisible by $L_{seg}$, we have $(k_2 - 1)L_{seg} < \tau < k_2 L_{seg}$ for some $k_2$. We set the learnable position embedding for decoder as $\mathbf{E}^{(dec)} \in \mathbb{R}^{k_2 \times D \times d_{model}}$ and input it to the decoder to get an output in shape of $\mathbb{R}^{k_2 L_{seg} \times D}$. Then the first $\tau$ steps of the output is used as $\mathbf{x}^{pred}_{T+1:T+\tau}$.

We conduct experiment on ETTm1 dataset to evaluate the effect of indivisible length. Results in Table 6 show that with padding mechanism, indivisible length does not degrade model performance, for both short-term prediction and long-term prediction.

### D.2  INCORPORATING COVARIATES

In the main text, we only use historical series $\mathbf{x}_{1:T}$ to forecast the future $\mathbf{x}_{T+1:T+\tau}$. In this section, we try to incorporate covariates $\mathbf{c}_{1:T+\tau}$ into Crossformer. We use a straightforward method: first embed the covariates into point-wise vectors $\{\mathbf{d}_1, \mathbf{d}_2, \ldots, \mathbf{d}_{T+\tau}\}$ like previous Transformer-based models do (Zhou et al., 2021; Wu et al., 2021a; Liu et al., 2021a). Then, merge the point-wise vectors into segment-wise vectors using learnable linear combination. Finally, add the segment-wise vectors to each dimension of the 2D vector array obtained by DSW embedding:

$$\mathbf{c}_t \to \mathbf{d}_t, 1 \le t \le T$$

$$\mathbf{d}_i^{(s)} = \sum_{0 < j \le L_{seg}} \alpha_j \mathbf{d}_{(i-1) \times L_{seg} + j}, \quad 1 \le i \le \frac{T}{L_{seg}} \tag{10}$$

$$\mathbf{h}_{i,d}^{cov} = \mathbf{h}_{i,d} + \mathbf{d}_i^{(s)}, \quad 1 \le i \le \frac{T}{L_{seg}}, \quad 1 \le d \le D$$

where $\to$ denotes embedding method for point-wise covariates. $\alpha_j, 1 \le j \le L_{seg}$ denotes learnable factors for linear combination. $\mathbf{d}_i^{(s)}$ denotes the segment-wise covariate embedding. $\mathbf{h}_{i,d}^{cov}$ denotes the

Table 7: MSE and MAE evaluation of Crossformer without/with covariates on ETTh1 dataset.

| Models | Crossformer | | Crossformer+Cov | |
|---|---|---|---|---|
| Metric | MSE | MAE | MSE | MAE |
| 24 | 0.305 | 0.367 | 0.308 | 0.368 |
| 48 | 0.352 | 0.394 | 0.358 | 0.399 |
| 168 | 0.410 | 0.441 | 0.412 | 0.440 |
| 336 | 0.440 | 0.461 | 0.438 | 0.465 |
| 720 | 0.519 | 0.524 | 0.522 | 0.531 |

embedded vector with covariate information for the $i$-th segment in dimension $d$, where $\mathbf{h}_{i,d}$ is the embedded vector obtained from DSW embedding in the main text. The processing for the input of the decoder is similar, the segment-wise covariate embedding is added to the position embedding for decoder, i.e. $\mathbf{E}^{(dec)}$.

We conduct experiments on ETTh1 dataset to evaluate the effect of covariates. Hour-of-the-day, day-of-the-week, day-of-the-month and day-of-the-year are used as covariates. Results in Table 7 show that incorporating covariates does not improve the performance of Crossformer. The possible reason is this straightforward embedding method does not cooperate well with Crossformer. Incorporating covariates into Crossformer to further improve prediction accuracy is still an open problem.

