# OpenReview forum: "Crossformer: Transformer Utilizing Cross-Dimension Dependency for Multivariate Time Series Forecasting"
_ICLR.cc/2023/Conference — ICLR 2023 notable top 5%_

### Official Review · Reviewer_74U5 · 2022-10-19

**Confidence:** 4
**Correctness:** 3
**Technical Novelty And Significance:** 2
**Empirical Novelty And Significance:** 3
**Recommendation:** 8

**Clarity, Quality, Novelty And Reproducibility:**

This paper was written in high quality and clarity, the figures are very illustrative and extensive experiments were conducted to compare the forecasting accuracy, memory occupation and computation time against multiple baselines. Moreover, ablation study was well-designed to cover the choice of multiple hyper-parameters. The paper is also very clear in presenting how the projections and layers are applied to each input vector in formulas.



Multivariate time series forecasting is a classic problem, and previous work has explored Transformer on long cross-time dependency, and CNN, GNN on cross-dimension dependency. This paper's originality is to use two transfomer's multi-head self attention layers to capture dependency in both dimension and time axis, and proposed corresponding segmented embedding, two stage attention layer and hierarchical encoder-decoder to get it work. Therefore, the originality is kind of limited.

**Details Of Ethics Concerns:**

N/A, there's no ethics conern in this paper.

**Strength And Weaknesses:**

This paper is well organized with comprehensive introduction into the network details and extensive experiment results. The setting of multivariate time series forecasting was well-introduced. The methodology part clearly presented how the initial input was segmented and fed into the 2-stage attention layer to capture cross-time and cross-dimension dependency. The figures are very helpful in understanding how the proposed modules work. The experiment part covers quite some baseline models and datasets, demonstrating the method's superior performance. The rich ablation study covers multiple aspects of experiment settings, including padding, covariants, hyper-parameter selection and architecture designs. These experiment results proved that the modules introduced are critical to the final performance. Moreover, the author(s) analyzed the time complexity of the encoder & decoder, and further benchmarked the memory occupation and computation time of each baseline model against CrossFormer. The very detailed comparison convinced me of the model's effectiveness.



On the other hand, the novelty of the paper is kinda limited because the problem of MTS forecasting and use of cross-dimension dependency was explored before, and Transformer was kind of the de facto in this setting. I also have a few concerns or questions as follows:

1. In Figure 1, it's interesting to see the pattern demonstrated in the figure(a). Can you please further elaborate how that figure indicates the MTS data tends to be segmented?

2. All the segments seem to be formed based on their location and segment length. Can you please explain how the segments align with the meaningful "segments" for forecasting in MTS data?

3. In section 3.2, the author(s) proposed the design of cross-dimension stage after cross-time stage. Is there any performance comparison between the current design with the other way around?

4. In sections 3.2, is there any performance comparison on the impact of using `LayerNorm` and skip connection?

5. It's quite surprising that covariates don't improve the performance in ablation study, can you share the covariates used in your experiments?

6. In the Hierarchical encoder decoder, the final results are summarized over each layer's output, is there any difference in the impact of each layer on the final results?



Minor suggestions:

7. In section 3, the author(s) can mention the covariants are not used earlier in the paper to avoid confusion.

8. In section 3.2, the author(s) used the notation of `L` to represent the number of segments, where $\frac{T}{L_{seg}}$ was also used in complexity analysis. They could be unified if there's no difference in their meanings.



**Summary Of The Paper:**

This paper proposed `CrossFormer`, a Transformer based neural network for multivariate time series (MTS) forecasting. The author(s) claimed that cross-dimension dependency was not well utilized and developed a method to explicitly explore and utilize cross-dimension dependency for MTS forecasting. In order to achieve that, the authors employed `Dimension-Segment-Wise` embedding to aggregate segment based embedding on the time axis, and designed a two stage attention layer using multi-head self attention and MLP to capture cross-time and cross-dimension dependency of the embedded input. A hierarchical encoder-decoder is further introduced to utilize the information at different scales. In this way, long term dependency between different time steps and dimensions can be captured. The author(s) compared the accuracy of the method on six real-world datasets and demonstrated superior results in most of the tasks. They also analyzed the theoretical time complexity of the method to be $O(\frac{D}{L_{seg}^2}{T^2})$ for encoder, $O(\frac{D}{L_{seg}^2}{(\tau(\tau+T))})$ for decoder, which performed better in terms of memory occupation and computation time in empirical experiments.


**Summary Of The Review:**


To summarize, this paper was written in high quality. It proposed `CrossFormer`, a Transformer based neural network, to capture cross-time and cross-dimension dependency, and achieved state-of-the-art performance in most of the multivariate time series forecasting tasks benchmarked, especially in long time series. The experiments were extensive and covered aspects of accuracy, memory occupation, computation time. The ablation study was comprehensive and proved the effectiveness of architectures proposed and hyper-parameters selected. There's a bit of a lack of novelty and this method may be hard to extend to other areas of representation learning. However, the insights from cross-dimension dependency and new design of 2-stage attention layers are very important. Therefore, I still recommend this paper to be accepted.

---

> ### Author Response · Authors · 2022-11-11
> **Response to Reviewer 74U5**
>
> Thank you for your time in reviewing our paper and the detailed comments. The followings are our responses to your specific questions:
> > ***Q1: Explanation of the Segmentation Pattern***
>
> The i-th row of the self-attention score map in Figure 1(a) can be roughly understood as computing the representation vector of step i, how the attention is distributed over the entire sequence. From the perspective of each row, We can see that close points in the time domain receive similar attention. This means to get a vector of step i, we can calculate the point-to-segment attention instead of the point-to-point attention. Moreover, notice that close rows show similar attention patterns, showing that we can further calculate the segment-to-segment attention to get representation vectors of segments instead of points.
>
> > ***Q2: How the Embedded Segments Align with the Meaningful "Segments"?***
> >
> The main purpose of DSW embedding is to divide the whole sequence into some basic elements for depend, or in other words atomic segments. Then these atomic segments are input into the encoder-decoder to capture dependency for forecasting. Therefore, the encoder-decoder will select meaningful atomic segments and perform alignment operations automatically. We do not need to define what is "meaningful" and align segments by hand through embedding.
>
> > ***Q3: Cross-Time First or Cross-Dimension First?***
> >
>
> We conducted experiments on ETTh1 dataset to compare these two designs. The MSE|MAE evaluation is shown in the following table. "Cross-Time First" stands for the original design in the paper where cross-dimension stage comes after cross-time stage and "Cross-Dimension First" represents the other way. The results show there is no big difference between these two designs. As long as the 2D vector array goes through these two stages one after the other, both cross-time and cross-dimension dependencies can be captured.
>
> | $\tau$ | Cross-Time First        | Cross-Dimension First |
> | ------ | ----------------------- | --------------------- |
> | 24     | 0.305&#124;0.367  |	0.307&#124;0.370 |
> | 48     | 0.352&#124;0.394  |	0.363&#124;0.411 |
> | 168    | 0.410&#124;0.441  |	0.407&#124;0.438 |
> | 336    | 0.440&#124;0.461  |	0.441&#124;0.462 |
> |    720    |      0.519&#124;0.524  |	0.518&#124;0.520           |
>
> > ***Q4: Ablation Study of LayerNorm and Skip Connection***
> >
>
> Skip connection and LayerNorm are widely used in Transformer-based MTS forecasting models, thus we use them as default. Here we perform ablation studies to investigate their impact on forecasting accuracy. The MSE|MAE comparison on ETTh1 dataset is shown in the following table. Results show that Skip connection constantly improves forecasting accuracy no matter whether LayerNorm is used or not. As for Layernorm, if Skip connection is not used, Layernorm helps to reduce forecasting error for long-term prediction ($\tau \ge 168$); if Skip connection is used, with or without LayerNorm gets similar performance. To get stable and accurate predictions, we use both of these two techniques in our Crossformer.
>
> | $\tau$ | Skip&cross;  LayerNorm&cross; | Skip&check;  LayerNorm&cross; | Skip&cross;  LayerNorm&check; | Skip&check;  LayerNorm&check; |
> | ----------- | ----------------------------- | ----------------------------- | ----------------------------- | ----------------------------- |
> | 24          | 0.331&#124;0.381                   | 0.308&#124;0.364                   | 0.337&#124;0.385                   | 0.305&#124;0.367                   |
> | 48          | 0.370&#124;0.403                   | 0.349&#124;0.396                   | 0.383&#124;0.414                   | 0.352&#124;0.394                   |
> | 168         | 0.687&#124;0.601                   | 0.413&#124;0.443                   | 0.515&#124;0.523                   | 0.410&#124;0.441                   |
> | 336         | 0.746&#124;0.631                   | 0.448&#124;0.463                   | 0.572&#124;0.557                   | 0.440&#124;0.461                   |
> | 720         | 0.811&#124;0.686                   | 0.524&#124;0.530                   | 0.618&#124;0.588                   | 0.519&#124;0.524
>
> > ***Q5: Impact of Covariates***
>
> We use same covariates as Informer[1], which are hour-of-the-day, day-of-the-week, day-of-the-month and day-of-the-year.
>
> Our main paper focuses on forecasting the future using only the past without covariates. And in Appendix D.2 we attempt to incorporate covariates into our design through embedding in a straightforward way: linear combination. The possible reason for the non-improvement is this simple embedding method does not cooperate well with Crossformer. Maybe with special and careful design, covariates help to improve prediction accuracy and we leave it as a future work.

---

> > ### Author Response · Authors · 2022-11-11
> > **Response to Reviewer 74U5 (Cont)**
> >
> > > ***Q6: The Impact of Predictions at Different Scales***
> > >
> > We visualize the hierarchical prediction pattern of the HED in $\underline{\text{Figure 8 of the Appendix}}$ of the newly submitted version. Roughly speaking, the top prediction layer (coarsest scale), captures the low-frequency general trend and periodic pattern of the future value. By adding predictions at finer scales, finer high-frequency patterns are added and the prediction gets closer to the ground truth curve.
> >
> > > ***Response to Minor suggestions***
> > - We have moved the descriptions about covariates forward to the beginning of the Methodology Section in the newly submitted version.
> > - We use segment merging in the HED, which means that the numbers of segments input to TSA at different scales are different: initially, $L = \frac{T}{L_{seg}}$; at a higher scale, $L = \frac{T}{2L_{seg}}$, etc. Therefore we use $L$ instead of $\frac{T}{L_{seg}}$ in Section 3.2 for consistency.
> >
> >
> > [1] Haoyi Zhou, Shanghang Zhang, Jieqi Peng, Shuai Zhang, Jianxin Li, Hui Xiong, and Wan Zhang. Informer: Beyond efficient transformer for long sequence time-series forecasting. In AAAI, 2021.

---

> > > ### Comment · Reviewer_74U5 · 2022-11-18
> > > **Response to the Authors**
> > >
> > > Thanks for the very detailed response, I'm happy to raise my score to 8 and recommend this paper to be accpeted as my questions are addressed.

---

### Official Review · Reviewer_SiSN · 2022-10-24

**Confidence:** 3
**Correctness:** 3
**Technical Novelty And Significance:** 3
**Empirical Novelty And Significance:** 3
**Recommendation:** 8

**Clarity, Quality, Novelty And Reproducibility:**

The paper is generally well-written and the main contributions are quite clear. The experimental results demonstrate the effectiveness of the proposed method.

**Strength And Weaknesses:**

[+] The idea for utilizing cross-dimension attention in multivariate time-series data seems novel and interesting. The reduced time complexity is also an advantage.

[+] Its effectiveness is validated through extensive experiments across six datasets. Sensitivity analysis on hyperparameter setting and ablation study is also provided.

[-] In computational efficiency analysis (Section 4.5), it would be better if the authors can also provide a comparison of actual execution time.

[-] In the ablation study (Section 4.3), the MSE of the TSA layer with and without the routers should be compared to validate the proposed router mechanism.

[-] The authors mention that HED decreases the performance when the prediction length is short, and the possible reason is the information on different scales is helpful to long-term prediction. However, when it is combined with TSA, short-length prediction shows good performance. The explanation of how TSA overcomes the shortcoming of HED should be addressed.


**Summary Of The Paper:**

This paper presents Crossformer, a Transformer-based model for multivariate time series (MTS) forecasting, especially focusing on cross-dimension dependency, which has not been widely investigated so far. They propose Dimension-Segment-Wise (DSW) embedding and Two-Stage Attention (TSA) layer to address this issue and further build a Hierarchical Encoder-Decoder (HED) to combine the information at different scales for final forecasting.

**Summary Of The Review:**

The paper is well-written overall. I think the contribution and novelty of the proposed work are clearly shown.

---

> ### Author Response · Authors · 2022-11-11
> **Response to Reviewer SiSN**
>
> We appreciate your valuable comments and thank you for identifying the novelty and extensive experimental results of our paper. The following are our responses to your specific questions and concerns:
>
> > ***Q1: Actual Execution Time Evaluation***
>
> The running speed evaluation is presented in $\underline{\text{Section B.4 of the Appendix}}$, and we have added a hint about this at the end of Section 4.5 of the main text. Results are consistent with that of memory occupation: Crossformer achieves the best computation speed compared with other Transformer-based models under different input lengths. And DSW+TSA+HED are faster than other ablation versions as the number of dimensions $D$ increases, indicating our designs help to reduce complexity.
>
> > ***Q2: Ablation Study of the Router Mechanism***
>
> We propose the router mechanism mainly to reduce the computation complexity when $D$ is large. And the efficiency evaluation in Figure 4(d) of the main paper shows it does make Crossformer more efficient.
>
> Per the request, here we evaluate its impact on forecasting accuracy. The MSE|MAE evaluation is shown in the following table. Results show that for short term prediction ($\tau \le 168$), the performances of TSA(w/o Router) and TSA are similar, no matter whether HED is used or not. For long term prediction, the router mechanism slightly improves the prediction accuracy. The possible reason is that we set seperate routers for each time step $i$, which makes the cross-dimension operation more flexible: differnet time steps can act differently. This design helps capture long-term dependency that varies over time.
>
> | $\tau$ | Transformer      | DSW              | DSW+TSA(w/o Router) | DSW+TSA          | DSW+HED          | DSW+TSA(w/o Router)+HED | DSW+TSA+HED      |
> | ------ | ---------------- | ---------------- | ------------------- | ---------------- | ---------------- | ----------------------- | ---------------- |
> | 24     | 0.620&#124;0.577 | 0.373&#124;0.418 | 0.320&#124;0.376    | 0.322&#124;0.373 | 0.406&#124;0.454 | 0.311&#124;0.375        | 0.305&#124;0.367 |
> | 48     | 0.692&#124;0.671 | 0.456&#124;0.479 | 0.356&#124;0.396    | 0.365&#124;0.403 | 0.493&#124;0.512 | 0.363&#124;0.406        | 0.352&#124;0.394 |
> | 168    | 0.947&#124;0.797 | 0.947&#124;0.731 | 0.487&#124;0.493    | 0.473&#124;0.479 | 0.614&#124;0.583 | 0.416&#124;0.444        | 0.410&#124;0.441 |
> | 336    | 1.094&#124;0.813 | 0.969&#124;0.752 | 0.585&#124;0.564    | 0.553&#124;0.534 | 0.788&#124;0.676 | 0.487&#124;0.499        | 0.440&#124;0.461 |
> | 720    | 1.241&#124;0.917 | 1.086&#124;0.814 | 0.665&#124;0.615    | 0.636&#124;0.599 | 0.841&#124;0.717 | 0.540&#124;0.542        | 0.519&#124;0.524 |
>
> > ***Q3: Why HED cooperates well with TSA?***
>
> HED is designed to cooperate with TSA. If we use DSW without TSA, to capture dependency among segments, the Self-Attention layer treats time and dimension axes equally and there is no difference between segments in the same dimension or other dimensions. While HED only merges adjacent segments along the time axis in the same dimension, which breaks the homogeneity of the previous operation. Different from the Self-Attention layer, the TSA layer treats the two axes differently, which matches the heterogeneous merging operation in HED. Moreover, the above table shows that HED also cooperates well with TSA (w/o Router). This also confirms our conclusion.

---

### Official Review · Reviewer_Zk9T · 2022-10-25

**Confidence:** 4
**Correctness:** 3
**Technical Novelty And Significance:** 2
**Empirical Novelty And Significance:** 2
**Recommendation:** 5

**Clarity, Quality, Novelty And Reproducibility:**

Clarity: fair
Quality: fair
Novelty: limited
Reproducibility: limited

**Strength And Weaknesses:**

Strength: Dimension information is an interesting topic in the time series. This paper proposes Crossformer, which performs cross-time and cross-dimension dependency. The experiment results are comprehensive.

Weaknesses:
(i) Efficiency is my main concern about this paper. Since 2-D attention is performed in Crossformer, the complexity is much more than in the previous 1-D attention method. Can authors provide some metric of efficiency (i.e., FLOPs, number of parameters, time) in Table 1?
(ii) Interestingly, there is another paper (https://openreview.net/forum?id=GpW327gxLTF) that investigates the dependency between dimensions. But their conclusion may be different from yours. For example, the dependency between ILI is significant compared with other data sets. However, in Table 1, Crossformer is not the best compared with other baselines on ILI. Can you explain that?
(iii) The extension from 1-D to 2-D is somehow trivial in time series fields. Thus, the novelty of the paper is weak in my view.


**Summary Of The Paper:**

This paper proposes Crossformer, a Transformer-based model utilizing cross-dimension dependency for Multivariate time series forecasting. Besides the time information, Crossformer considers dependency between dimension information. Specifically, Dimension-Segment-Wise (DSW) embedding and Two-Stage Attention (TSA) layer are proposed to efficiently capture both cross-time and cross-dimension dependency. Extensive experimental results show the effectiveness of Crossformer.

**Summary Of The Review:**

This paper proposes a cross-time and cross-dimension model - Crossformer. However, efficiency is my main concern since Crossformer should perform 2-D attention and the paper doesn't provide any efficiency metric. Thus, I think the paper is not much practical.

---

> ### Author Response · Authors · 2022-11-11
> **Response to Reviewer Zk9T**
>
> Thanks for your time and effort in reviewing our paper. We are pleased to see that you think the topic of utilizing cross-dimension dependency for MTS forecasting is interesting and important. However, we guess that there are some misunderstandings about the efficiency and novelty of our work, which we try to clarify in our following responses to specific questions and concerns:
>
> >***Q1: Efficiency and Complexity***
>
> In general, the additional overhead is caused by the explicit utilization of cross-dimension dependency in Crossformer. Compared with previous Transformer-based models which **omit the cross-dimension dependency**, the explicit utilization will introduce additional computational overhead w.r.t to $D$, where $D$ is the number of dimensions of the MTS data.
>
> We have made non-trivial efforts to reduce the additional complexity caused by cross-dimension dependency to make Crossformer practically usable: 1) Directly apply Self-Attention to the 2D input has the complexity of $O(D^2T^2)$; 2) Using DSW embedding reduces the complexity to $O(\frac{D^2}{L_{seg}^2}T^2)$; 3) Using TSA layer without the router mechanism reduce it to $O(\frac{D}{L_{seg}^2}T^2+\frac{D^2}{L_{seg}}T)$; 4) Finally, using TSA layer with the router mechanism reduces it to $O(\frac{D}{L_{seg}^2}T^2)$. Theoretically, if a model maintains the information of all dimensions in the latent space like our Crossformer, the complexity linearized to $D$ is the best it can achieve, as it at least needs to go through all dimensions.
>
> In $\underline{\text{Figure 4(c) of the main paper}}$ and $\underline{\text{Figure 9(a) of the Appendix}}$, we carefully align hyper-parameters of Crossformer with other Transformer-based models and evaluate their memory occupation and running speed w.r.t input length $T$ on ETTh1 dataset ($D = 7$). The result shows that Crossformer achieves the best efficiency within the tested length range. As for complexity w.r.t $D$, considering baseline Transformer-based models do not model cross-dimension dependency explicitly, we compare the efficiency of Crossformer with its ablation versions in $\underline{\text{Figure 4(d) of the main paper}}$ and $\underline{\text{Figure 9(b) of the Appendix}}$. The results show that our design greatly reduces the complexity and makes it possible to process high-dimensional data.
>
> Per the request, here we show the memory cost (in GB) and training speed (in second/batch) of Transformer-based models and MTGNN to achieve the performance in Table 1 of the main paper. It should be noticed that many hyper-parameters, such as input length, number of layers in encoder and decoder, etc. have influence on the efficiency and they were adjusted to get the best prediction accuracy. Therefore, the following comparison should be taken as a rough reference.
>
> The $\left(\text{memory occupation (in GB) | training speed (in s/batch)}\right)$ on low-dimensional datasets ETTh1 ($D = 7$) and WTH ($D = 12$) are shown in the following tables. **We can see that in most settings, Crossformer requires less memory and runs faster than other Transformer-based models.**
> | ETTh1 ($D = 7$) | Transformer        | Informer           | Autoformer         | FEDformer          | MTGNN              | Crossformer        |
> | ----- | ----- | ---- | ----- | ------ | ----- | ------ |
> | $\tau = 24$     | 0.4890&#124;0.0204 | 0.4934&#124;0.0378 | 1.1220&#124;0.0600 | 0.4398&#124;0.1077 | 0.1462&#124;0.0426 | 0.7849&#124;0.0686 |
> | $\tau = 48$     | 0.6865&#124;0.0317 | 0.7763&#124;0.0440 | 1.2686&#124;0.0590 | 0.4810&#124;0.1127 | 0.1464&#124;0.0422 | 0.9495&#124;0.0695 |
> | $\tau = 168$    | 1.7685&#124;0.0836 | 2.0026&#124;0.0902 | 2.0962&#124;0.0828 | 0.6797&#124;0.1302 | 0.2554&#124;0.0472 | 0.9576&#124;0.0671 |
> | $\tau = 336$    | 2.5507&#124;0.1117 | 3.0557&#124;0.1089 | 3.1914&#124;0.1240 | 1.6472&#124;0.2417 | 0.6050&#124;0.0507 | 1.2313&#124;0.0697 |
> | $\tau = 720$    | 7.3777&#124;0.2860 | 6.1354&#124;0.2188 | 6.0076&#124;0.2189 | 2.2381&#124;0.3106 | 1.0899&#124;0.0860 | 1.9043&#124;0.1014 |
>
> | WTH ($D = 12$) | Transformer         | Informer           | Autoformer         | FEDformer          | MTGNN              | Crossformer        |
> | ----- | ------ | ------ | ------ | ----- | ----- | ----- |
> | $\tau = 24$    | 2.2226&#124;0.1334  | 1.8399&#124;0.1197 | 0.5453&#124;0.0436 | 0.4405&#124;0.1076 | 0.2377&#124;0.0417 | 0.6141&#124;0.0727 |
> | $\tau = 48$    | 1.8577&#124;0.0470  | 1.6733&#124;0.0522 | 0.6816&#124;0.0507 | 0.4819&#124;0.1119 | 0.2381&#124;0.0455 | 0.5097&#124;0.0775 |
> | $\tau = 168$   | 5.0866&#124;0.2218  | 3.0908&#124;0.1904 | 1.4348&#124;0.0618 | 0.9177&#124;0.1672 | 0.4084&#124;0.0478 | 0.9586&#124;0.0713 |
> | $\tau = 336$   | 13.2690&#124;0.4470 | 5.3654&#124;0.3127 | 3.1939&#124;0.1283 | 1.6509&#124;0.2429 | 0.9214&#124;0.0715 | 1.4023&#124;0.0811 |
> | $\tau = 720$   | 22.9143&#124;0.7548 | 8.7855&#124;0.4854 | 8.8988&#124;0.3444 | 2.2473&#124;0.3128 | 1.7825&#124;0.1274 | 2.9413&#124;0.1461 |

---

> > ### Author Response · Authors · 2022-11-11
> > **Response to Reviewer Zk9T (Cont)**
> >
> > The $\left(\text{memory occupation (in GB) | training speed (in s/batch)}\right)$ on high-dimensional dataset Traffic ($D = 862$) are shown in the following table. To avoid the out-of-memory problem on a single GPU with 48GB memory, MTGNN can only use a 1-layer encoder, but Crossformer still uses a 3-layer encoder. Moreover, we have to reduce the batch size to 16 for MTGNN when $\tau = 336,720$, but other models always use the batch size of 32. We can see that without explicit utilization of cross-dimension dependency, previous Transformer-based models are more efficient than Crossformer. **But compared with MTGNN, which also takes cross-dimension dependency into consideration explicitly, Crossformer is more efficient.** Moreover, the MSE and MAE of MTGNN and Crossformer are much lower than other Transformers on this dataset.
> > | WTH ($D = 862$) | Transformer         | Informer           | Autoformer         | FEDformer          | MTGNN              | Crossformer        |
> > | -------------- | ------------------- | ------------------ | ------------------ | ------------------ | ------------------ | ------------------ |
> > | $\tau = 24$    | 0.3927&#124;0.0262 |	0.3934&#124;0.0281|	0.8021&#124;0.0458|	4.2202&#124;0.2014|	7.7571&#124;0.5067|	5.1694&#124;0.3104 |
> > | $\tau = 48$    | 0.8294&#124;0.0457|	0.4729&#124;0.0329|	1.4379&#124;0.0710|	4.7069&#124;0.4511|	7.7730&#124;0.5281|	7.0761&#124;0.4101 |
> > | $\tau = 168$   | 1.2491&#124;0.0671|	1.1677&#124;0.0943|	2.3792&#124;0.1089|	4.9032&#124;0.6691|	22.3130&#124;1.6202|	12.8833&#124;0.7000 |
> > | $\tau = 336$   | 4.3073&#124;0.2040|	1.7032&#124;0.1407|	3.6265&#124;0.1697|	7.5981&#124;1.2813|	24.2138&#124;1.9593 (batch_size=16)|	27.5025&#124;1.3920 |
> > | $\tau = 720$   | 4.3389&#124;0.2114|	3.2196&#124;0.2324|	9.8062&#124;0.4384	|9.4646&#124;1.8791|	24.7323&#124;2.0466 (batch_size=16)|	37.5490&#124;1.8455
> >
> > >***Q2: Performance on ILI and Relation to UvsM Paper***
> > >
> > We conjecture the reason why Crossformer is not the best compared with other baselines on ILI is that the size of ILI is much smaller than other datasets: ILI is a **weekly recorded** dataset and its **overall length is 966**. While other datasets are recorded **every hour or every 15 minutes** and all of them are **longer than 10,000**. Autoformer and FEDformer introduce the prior knowledge such as sequence decomposition into the network structure, while Crossformer needs enough data to surpass such inductive bias. Therefore, Autoformer and FEDformer perform better when the data is limited. This is consistent with the claim in the UvsM paper that ''...we are cautious to draw conclusions from this due to the exceptionally small size of the dataset. Illness (ILI) has a times series length 70 times shorter than the longest, ETT'' and ''more data is required'' for multivariate models.
> >
> > Moreover, from the perspective of model's representation capability, our Crossformer is more general and can cover the univariate setting in the UvsM paper: assuming there is no cross-dimension (or inter-dimensional in UvsM paper) dependency in the MTS data, and thus the cross-dimension stage of Crossformer just copies the output of cross-time stage via skip connection. In cross-time stage, all dimensions share the same MSA layer, thus the prediction of each dimension is made by the same attention model. In this way, the Crossformer degenerates to the univariate setting of the UvsM paper. This suggests Crossformer is a general model that decides whether to extract cross-dimension dependency or not automatically based on the input data.

---

> > > ### Author Response · Authors · 2022-11-11
> > > **Response to Reviewer Zk9T (Cont)**
> > >
> > > > ***Q3: Novelty***
> > > >
> > > **Firstly, the extension from 1D to 2D is not so straightforward for MTS forecasting Transformers.** As Transformers were first proposed in NLP to process 1D sequential data, most previous works (e.g. Informer[1], Autoformer[2], Pyraformer[3], FEDformer[4]) follow this setting by embedding the MTS data into a 1D sequence and capturing the cross-time dependency among different time steps. The ablation study in $\underline{\text{Section 4.3 of the main paper}}$ shows that extension from 1D to 2D is an effective idea: from Transformer to DSW, the only difference is the embedding methods and the model architectures are the same (we flatten the 2D vector array into a 1D sequence to be input into the same Transformer). Results show that explicitly utilizing cross-dimension dependency constantly improves model performance, even in such a naive way.
> > >
> > > **Secondly, extension from 1D to 2D causes additional technical challenges to handle via new designs.** The TSA layer is proposed to deal with the heterogeneity of time and dimension axis. The router mechanism is proposed to reduce the complexity of cross-dimension stage so that our Crossformer can process high-dimensional data. The HED is used to extract information in different scales to improve forecasting accuracy.
> > >
> > > **Finally, our work can inspire future research on MTS forecasting with Transformers that besides temporal dependency, cross-dimension dependency should also be taken into consideration carefully for model design.**
> > >
> > > [1] Haoyi Zhou, Shanghang Zhang, Jieqi Peng, Shuai Zhang, Jianxin Li, Hui Xiong, and Wan Zhang. Informer: Beyond efficient transformer for long sequence time-series forecasting. In AAAI, 2021.
> > >
> > > [2] Haixu Wu, Jiehui Xu, Jianmin Wang, and Mingsheng Long. Autoformer: Decomposition transformers with auto-correlation for long-term series forecasting. In NeurIPS, 2021.
> > >
> > > [3] Shizhan Liu, Hang Yu, Cong Liao, Jianguo Li, Weiyao Lin, Alex X Liu, and Schahram Dustdar. Pyraformer: Low-complexity pyramidal attention for long-range time series modeling and forecasting. In ICLR, 2021.
> > >
> > > [4] Tian Zhou, Ziqing Ma, Qingsong Wen, Xue Wang, Liang Sun, Rong Jin. FEDformer: Frequency Enhanced Decomposed Transformer for Long-term Series Forecasting. In ICML, 2022.

---

> ### Author Response · Authors · 2022-12-01
> **Looking Forward to Your Feedback**
>
> Dear Reviewer Zk9T,
>
> Thank you once again for reviewing our paper.
>
> In our early response, we have included a detailed analysis of the efficiency and novelty of our model. Moreover, we have provided the efficiency evaluation results pre your request.
>
> We would like to know whether our response has addressed your concerns. If you have any further questions, please do not hesitate to let us know, so that we can respond to them timely.
>
> Sincerely, Authors

---

### Official Review · Reviewer_pZRS · 2022-10-25

**Confidence:** 4
**Clarity, Quality, Novelty And Reproducibility:** This study is clear and pretty novel.
**Correctness:** 4
**Technical Novelty And Significance:** 3
**Empirical Novelty And Significance:** 3
**Recommendation:** 8

**Strength And Weaknesses:**

This manuscript is well written and organized very well. Also  the studied topic is very interesting as it is usually the case that for MTS the cross dimension dependencies is overlooked.

The authors argue that as “For MTS, a single value at a step alone provides little information”, embedding in a way that “nearby points over time form a segment” could be beneficial. This argument seems interesting and valid as it is used and illustrated nicely (In Figure 1) by the authors.

In my opinion this is an interesting study.   I think you can complete your great work by comparing your method with “Long-Range Transformers for Dynamic Spatiotemporal Forecasting” which you can find in “https://arxiv.org/pdf/2109.12218.pdf” and also include the very simple baseline that is mentioned in this study “https://arxiv.org/pdf/2205.13504.pdf”.


**Summary Of The Paper:**

This study addresses Multivariate time series forecasting using a transformer-based model. Namely, the authors presented Crossformer, a transformer based model which not only takes into account the temporal sequence, but also the cross dimension dependencies among the variables.
The inputs for Crossformer, is embedded using Dimension-Segment-Wise (DSW) embedding in a 2d array then it is processed by using a two-stage attention layer that  supposedly captures both cross-time and cross-dimension dependencies.
In the cross-time stage,  a multi-head self-attention (MSA) is applied to each dimension and then an MLP is used to capture the dependency among the different segments in each dimension.
In the cross-dimension stage, as directly applying MSA for high dimensional datasets will not be affordable, the authors proposed using router mechanism to aggregate messages from all dimensions into a low dimension space.
Then to make the all-to-all connections,  the aggregated
messages are considered as key and value and the vectors of dimensions as query.
Finally, the authors used a hierarchical  encoder-decoder structure to use the extracted information in different scales.


**Summary Of The Review:**

In my opinion, this study is nicely done and by adding two mentioned comparison above will be completed enough to be published in this conference.

---

> ### Author Response · Authors · 2022-11-11
> **Response to Reviewer pZRS**
>
> Thank you for identifying the novelty of our work and providing us with these two interesting concurrent studies. Now, we have finished reading these two works and we are working hard to make the comparison. Since we need to review the code, align settings, adjust hyper-parameters, etc., we may need a few more days to get the quantitative evaluation results. We will come back to you soon once we get the results.

---

> > ### Comment · Reviewer_pZRS · 2022-12-01
> > **Thanks for trying to make your comparison comprehensive**
> >
> > I am interested to see the further experiments that you mentioned before, if you have them.

---

> > > ### Author Response · Authors · 2022-12-02
> > > **Comprehensive Comparison with Additional Models**
> > >
> > > We have already evaluated the performance of STformer[1] and DLinear[2] on the first three datasets (ETTh1, ETTm1, WTH). The comprehensive $\left(\text{MSE|MAE}\right)$ evaluation is shown in the following three tables. Here, we omit LSTMa, LSTnet, Transformer, and Informer as they are not competitive with other models. Bold indicates the best and underline indicates the second best.
> > > > ***ETTh1***
> > >
> > > | $\tau$ | MTGNN       | AutoFormer  | Pyraformer  | FEDformer   | STformer    | DLinear     | Crossformer |
> > > | ------ | ----------- | ----------- | ----------- | ----------- | ----------- | ----------- | ----------- |
> > > | 24     | 0.336&#124;0.393 | 0.439&#124;0.440 | 0.493&#124;0.507 | 0.318&#124;0.384 | 0.368&#124;0.411 | $\underline{0.312}$&#124;**0.355** | **0.305**&#124;$\underline{0.367}$ |
> > > | 48     | 0.386&#124;0.429 | 0.429&#124;0.442 | 0.554&#124;0.544 | **0.342**&#124;0.396 | 0.445&#124;0.465 | $\underline{0.352}$&#124;**0.383** | $\underline{0.352}$&#124;$\underline{0.394}$ |
> > > | 168    | 0.466&#124;0.474 | 0.493&#124;0.479 | 0.781&#124;0.675 | $\underline{0.412}$&#124;0.449 | 0.652&#124;0.608 | 0.416&#124;**0.430** | **0.410**&#124;$\underline{0.441}$ |
> > > | 336    | 0.736&#124;0.643 | 0.509&#124;0.492 | 0.912&#124;0.747 | 0.456&#124;0.474 | 1.069&#124;0.806 | $\underline{0.450}$&#124;**0.452** | **0.440**&#124;$\underline{0.461}$ |
> > > | 720       |0.916&#124;0.750|	0.539&#124;0.537|	0.993&#124;0.792|	0.521&#124;**0.515**|	1.0712&#124;0.817|	**0.512**&#124;$\underline{0.523}$|	$\underline{0.519}$&#124;0.524|
> > >
> > > > ***ETTm1***
> > >
> > > | $\tau$ | MTGNN       | AutoFormer  | Pyraformer  | FEDformer   | STformer    | DLinear     | Crossformer |
> > > | ------ | ----------- | ----------- | ----------- | ----------- | ----------- | ----------- | ----------- |
> > > | 24     | 0.260&#124;0.324|	0.410&#124;0.428|	0.310&#124;0.371|	0.290&#124;0.364|	0.278&#124;0.348|	$\underline{0.217}$&#124;**0.289**|	**0.211**&#124;$\underline{0.293}$|
> > > | 48     | 0.386&#124;0.408|	0.485&#124;0.464|	0.465&#124;0.464|	0.342&#124;0.396|	0.445&#124;0.458|	**0.278**&#124;**0.330**|	$\underline{0.300}$&#124;$\underline{0.352}$|
> > > | 96    | 0.428&#124;0.446|	0.502&#124;0.476|	0.520&#124;0.504|	0.366&#124;0.412|	0.420&#124;0.455|	**0.310**&#124;**0.354**|	$\underline{0.320}$&#124;$\underline{0.373}$ |
> > > | 288    | 0.469&#124;0.488|	0.604&#124;0.522|	0.729&#124;0.657|	$\underline{0.398}$&#124;0.433|	0.733&#124;0.597|	**0.369**&#124;**0.386**|	0.404&#124;$\underline{0.427}$ |
> > > | 672       |0.620&#124;0.571|	0.607&#124;0.530|	0.980&#124;0.768|	$\underline{0.455}$&#124;$\underline{0.464}$|	0.777&#124;0.625|	**0.416**&#124;**0.417**|	0.569&#124;0.528|
> > >
> > > > ***WTH***
> > >
> > > | $\tau$ | MTGNN       | AutoFormer  | Pyraformer  | FEDformer   | STformer    | DLinear     | Crossformer |
> > > | ------ | ----------- | ----------- | ----------- | ----------- | ----------- | ----------- | ----------- |
> > > | 24     | 0.307&#124;0.356|	0.363&#124;0.396|	$\underline{0.301}$&#124;$\underline{0.359}$|	0.357&#124;0.412|	0.307&#124;$\underline{0.359}$|	0.357&#124;0.391|	**0.294**&#124;**0.343**|
> > > | 48     | 0.388&#124;0.422|	0.456&#124;0.462|	$\underline{0.376}$&#124;0.421|	0.428&#124;0.458|	0.381&#124;$\underline{0.416}$|	0.425&#124;0.444|	**0.370**&#124;**0.411**|
> > > | 168    | 0.498&#124;0.512|	0.574&#124;0.548|	0.519&#124;0.521|	0.564&#124;0.541|	$\underline{0.497}$&#124;0.502|	0.515&#124;0.516|	**0.473**&#124;**0.494** |
> > > | 336    | $\underline{0.506}$&#124;$\underline{0.523}$|	0.600&#124;0.571|	0.539&#124;0.543|	0.533&#124;0.536|	0.566&#124;0.564|	0.536&#124;0.537|	**0.495**&#124;**0.515**|
> > > | 720       |**0.510**&#124;**0.527**|	0.587&#124;0.570|	0.547&#124;0.553|	0.562&#124;0.557|	0.589&#124;0.582|	0.582&#124;0.571|	$\underline{0.526}$&#124;$\underline{0.542}$|
> > >
> > > STformer outperforms Autoformer and Pyraformer on ETTh1 and ETTm1 when $\tau \le 96$ and achieves the second best on WTH when $\tau \le 168$. The basic idea of STformer is similar to Crossformer: both of them extend the 1-D attention to 2-D. However, STformer directly flattens the raw $T \times D$  2-D serirs into a 1-D sequence to be input to the Transformer. This straightforward method does not distinguish the time and dimension axes and is computationally inefficient. Therefore, besides the good performance for short-term prediction, results also show that STformer has difficulty in long-term prediction. While Crossformer uses the DSW embedding to capture local dependency and reduce computation complexity. The TSA layer with the router mechanism is proposed to deal with the heterogeneity of time and dimension axis and further improve efficiency.

---

> > > > ### Author Response · Authors · 2022-12-02
> > > > **Comprehensive Comparison with Additional Models (Cont)**
> > > >
> > > > DLinear is competitive with FEDFormer and our Crossformer on ETTh1; constantly outperforms all Transformer-based models including Crossformer on ETTm1 when $\tau \ge 48$; performs worse than MTGNN and our Crossformer on WTH. Considering the simplicity of DLinear, such results are very competitive.
> > > >
> > > > Authors of DLinear suggest that in time series modeling, we need to extract the temporal relations in an ordered set of continuous points, rather than capturing the semantic correlations among elements in the sequence. This is consistent with our argument that "For MTS, a single value at a step alone provides little information. While it forms informative pattern with nearby values in time domain." They also argue that Transformer-based models have difficulty in preserving ordering information because the attention mechanism is permutation-invariant and the absolute position embedding injected into the model is not enough for this order-sensitive task. We conducted detailed investigations and found that **relative position encoding** in NLP[3,4] and CV[5] may be a promising direction to overcome this shortcoming.
> > > >
> > > > It should be noticed that DLinear "does not model correlations among variates". Therefore, incorporating cross-dimension dependency into DLinear to further improve prediction accuracy is also a promising direction for future research. Moreover, our DSW embedding to enhance locality and HED to capture dependency at different scales can be potentially useful to further inspire and enhance the DLinear model.
> > > >
> > > >
> > > > References
> > > >
> > > > [1] Grigsby, Jake, Zhe Wang, and Yanjun Qi. Long-range transformers for dynamic spatiotemporal forecasting. arXiv preprint arXiv:2109.12218, 2021.
> > > >
> > > > [2] Zeng, Ailing, Muxi Chen, Lei Zhang, and Qiang Xu. Are Transformers Effective for Time Series Forecasting?. arXiv preprint arXiv:2205.13504, 2022.
> > > >
> > > > [3] Ke, Guolin, Di He, and Tie-Yan Liu. Rethinking Positional Encoding in Language Pre-training. In ICLR, 2020.
> > > >
> > > > [4] Dufter, Philipp, Martin Schmitt, and Hinrich Schütze. Position information in transformers: An overview. Computational Linguistics, 2022.
> > > >
> > > > [5] Wu, Kan, Houwen Peng, Minghao Chen, Jianlong Fu, and Hongyang Chao. Rethinking and improving relative position encoding for vision transformer. In ICCV, 2021.

---

### Public Comment · ~Grigory_Evko1 · 2023-02-04
**What about Github source code?**

Hi, great paper!
Can I ask when you plan to publish the source code for the article?

Also, suppose I have multi-channel time-series where each channel represents essentially different data, with different distributions. As far as I understand, the model described in the article does not imply the use of different data in the channels? Could you suggest modifications to use your model with multi-channel data with different distributions?

UPD
Forgot to mention that I'm referring to MS task type, where you have multi-channel input and single channel output.

Thanks!

---

> ### Author Response · Authors · 2023-02-12
> **Responses about Source Code and Modifications**
>
> Thank you for your interest. The source code will be publicly available along with the final camera-ready paper.
>
> One simple and efficient way to handle the multi-channel data with different distributions you mentioned is pre-processing. For example, you can conduct **normalization** to each dimension with its past mean and variance, which has already been done in our experiments.  Also, you can perform **season-trend decomposition** on the data and predict the two decomposed terms separately.
>
> As for your MS task, you need to modify the decoder, we can come up with two ways:  1) flatten the encoded 2D vector array into a 1D sequence to be used as keys and values of the multi-head self-attention; 2) use the same decoder as our original paper (i.e. each dimension will make a prediction of the target series), and sum over the predicted series of all dimensions to get the final result.

---

### Public Comment · ~Jingge_Xiao1 · 2023-02-10
**Would you like to publish the source code of Crossformer?**

Very interesting paper! Would you like to publish the source code?

---

> ### Author Response · Authors · 2023-02-12
> **The source code will be publicly available along with the final camera-ready paper.**
>
> Thank you for your interest in our work. The source code will be publicly available along with the final camera-ready paper.

---

### Decision · Program_Chairs · 2023-01-20

**Decision:**

Accept: notable-top-5%

**Justification For Why Not Higher Score:**

N/A

**Justification For Why Not Lower Score:**

All aspects of this paper are solid. The authors have answered all concerns, so I see no reason to give it a lower score.

**Metareview: Summary, Strengths And Weaknesses:**


The paper proposes a Transformer-based model that considers dependencies between variables, which the authors refer to as cross-dimension dependency, using 2-stage attention. The method is novel, the experiments are convincing and the study is well written. Reviewer pZRS identified concurrent work, and the authors were quick to run experiments to compare their method against those methods, showing Crossformer is better performing. Reviewer Zk9T pointed out the issue of efficiency, however, the authors responded with their complexity analysis, showing that Crossformer is efficient - I consider their arguments valid. While the reviewer did not respond, I'll consider this concern adequately addressed. The ablation studies in the paper were appreciated, as was the overall structure and writing of the paper. There was a question about whether cross-time or cross-dimension dependencies should be considered first. The authors responded with experiments showing that the method works well with either option.
Overall, this is an excellent paper and should be of great interest to the community.

**Note From Pc:**

if the above contains the word "oral" or "spotlight" please see: "oral" presentation means -> notable-top-5% and "spotlight" means -> notable-top-25%. As stated in our emails, we are disassociating presentation type from AC recommendations

**Summary Of Ac-Reviewer Meeting:**

N/A